# Fantastic Targets for Concept Erasure in Diffusion Models and Where To Find Them

Anh Bui[1], Thuy-Trang Vu[1], Long Vuong[1], Trung Le[1], Paul Montague[2],
Tamas Abraham[2], Junae Kim[2], and Dinh Phung[1]

[1]Monash University
[2]Defence Science and Technology Group, Australia

## Abstract

Concept erasure has emerged as a promising technique for mitigating the risk of harmful content generation in diffusion models by selectively unlearning undesirable concepts. The common principle of previous works to remove a specific concept is to map it to a fixed generic concept, such as a neutral concept or just an empty text prompt. In this paper, we demonstrate that this fixed-target strategy is suboptimal, as it fails to account for the impact of erasing one concept on the others. To address this limitation, we model the concept space as a graph and empirically analyze the effects of erasing one concept on the remaining concepts. Our analysis uncovers intriguing geometric properties of the concept space, where the influence of erasing a concept is confined to a local region. Building on this insight, we propose the Adaptive Guided Erasure (AGE) method, which *dynamically* selects optimal target concepts tailored to each undesirable concept, minimizing unintended side effects. Experimental results show that AGE significantly outperforms state-of-the-art erasure methods on preserving unrelated concepts while maintaining effective erasure performance. Our code is published at https://github.com/tuananhbui89/Adaptive-Guided-Erasure.

## 1 Introduction

The widespread accessibility of text-to-image generation models has introduced significant risks such as the generation of harmful content, copyright infringements, and biases due to the exposure of models to undesirable concepts during training. Initial attempts to address these concerns focus on dataset filtering during the training phase (StabilityAI, 2022), post-generating filtering (Rando et al., 2022) or inference guiding (Schramowski et al., 2023b). While dataset filtering often requires costly retraining, post-processing and inference-based methods can be easily bypassed (Yang et al., 2024). Recently, concept erasure (Bui et al., 2024b;a; Gandikota et al., 2023; Orgad et al., 2023; Zhang et al., 2023; Kumari et al., 2023), which aims to directly remove the concept from the model's parameters without the need for complete retraining, has emerged as a promising alternative.

Concept erasure methods can be broadly categorized into two approaches: output-based and attention-based. Output-based methods aim to neutralize the output associated with undesirable concepts (Bui et al., 2024b;a; Gandikota et al., 2023; Wu et al., 2024), while attention-based methods modify the attention scores of these concepts in the cross-attention layers of the model (Zhang et al., 2023; Orgad et al., 2023; Gandikota et al., 2024; Lu et al., 2024; Lyu et al., 2024). Despite their differences, both approaches share a common principle: mapping undesirable concepts to a fixed, generic target, such as "a photo" or an empty text prompt. Although these methods have shown success in erasing undesirable concepts, they do not consider how the choice of target concepts affects both the effectiveness of erasure and the preservation of benign concepts.

To address this limitation, we first model the concept space as a graph, where each node represents a concept, and the edge weights denote the impact of erasing one concept on another. We perform an empirical analysis using a newly curated evaluation dataset, NetFive, to understand the structure of the concept space. Our findings suggest that the concept impact space has a geometric structure,

characterized by a key property: *locality*—the impact of erasing one concept is localized in the concept space, i.e., it affects **strongly** only those concepts that are close to the erased one.

Furthermore, we explore different types of target concepts—synonyms, semantically related local concepts, and semantically distant concepts—and find that the choice of target significantly influences both erasure performance and the preservation of benign concepts. At the end, we identify the ideal target concept as the most affected by the change of the model parameters when the corresponding concept is erased, but must not be its synonym.

Building on these insights, we propose the Adaptive Guided Erasure (AGE) method, which automatically selects the optimal target concept for each erasure query by solving a minimax optimization problem. To further improve the richness of the target concept, we model it as a learned mixture of multiple single concepts, allowing us to search for the optimal target in continuous rather than discrete space. We evaluate the proposed AGE method on various erasure tasks, including object removal, Not-Safe-For-Work (NSFW) attribute erasure, and artistic style removal. Experimental results show that AGE significantly outperforms state-of-the-art concept erasure methods, achieving near-perfect preservation of benign concepts while effectively erasing undesirable ones.

Our main contributions can be summarized as follows. ❶ We present a novel empirical evaluation of the structure and geometric properties of the concept space, which provides new insights into concept erasure, such as the locality of the impact of erasing one concept on another. ❷ We analyze the impact of target concept selection on both erasure effectiveness and the preservation of benign concepts, identifying two key properties of desirable target concepts, i.e., closely related but not synonyms to the to-be-erased concept. ❸ Motivated by the analysis, we propose a novel adaptive method for selecting the optimal target concept for each erasure query satisfying the two key properties. ❹ Finally, we conduct extensive experiments demonstrating the effectiveness of our method and its ability to preserve benign concepts while erasing undesirable ones.

## 2 BACKGROUND

**Latent Diffusion Models.** Diffusion models, a recent class of generative models, have shown impressive results in generating high-resolution images (Ho et al., 2020; Rombach et al., 2022; Ramesh et al., 2021; 2022). In a nutshell, training a diffusion model involves two processes: a forward diffusion process where noise is gradually added to the input image $x_0 \sim p_{\text{data}}$, and a reverse denoising diffusion process where the model tries to predict a noise $\epsilon_t$, which is added in the forward process. The model is trained by minimizing the difference between the true noise $\epsilon$ and the predicted noise $\epsilon_\theta(x_t, t)$ at diffusion step $t$ parameterized by the denoising model $\theta$. With an intuition that semantic information that controls the main concept of an image can be represented in a low-dimensional space, Rombach et al. (2022) proposed a diffusion process operating on the latent space of a pre-trained encoder $\mathcal{E}$ which compresses the input data $x_t$ into low-dimensional latent representation $z_0 = \mathcal{E}(x)$. The objective function of the latent diffusion model as follows:

$$\mathcal{L} = \mathbb{E}_{z_0 \sim \mathcal{E}(x), x \sim p_{\text{data}}, t, \epsilon \sim \mathcal{N}(0, \mathbf{I})} \left\| \epsilon - \epsilon_\theta(z_t, t) \right\|_2^2 \tag{1}$$

**Concept Erasing.** Given a textual description in a set of undesirable concepts $c_e \in \mathbf{E}$, the concept erasing problem aims to remove this concept from a pretrained text-to-image diffusion model $\epsilon_\theta(z_t, \tau(c), t)$, typically via finetuning to obtain a benign output $\epsilon_{\theta'}(z_t, \tau(c), t)$ from *sanitized* model $\epsilon_{\theta'}$ parameterized by $\theta'$. For the remainder of this paper, because all outputs are from the same time step $t$ and latent vector $z_t$, we omit the time step $t$ and latent vector $z_t$ for the sake of simplicity, i.e., $\epsilon_\theta(\tau(c))$ and $\epsilon_{\theta'}(\tau(c))$. While proposed in different forms, previous works (Gandikota et al., 2023; Orgad et al., 2023; Gandikota et al., 2024) share a common principle to map a to-be-erased concept $c_e$ to a target neutral concept $c_t$. The target concept $c_t$ can be either a generic concept, such as "a photo" or a null concept, such as an empty text prompt. It is typically predefined and fixed for all undesirable concepts $c_e \in \mathbf{E}$. To preserve the model's performance on other concepts, an additional term $L_2$ is added to ensure that the output of the neutral concept remains unchanged. More specifically, the erasing objective can be formulated as follows:

$$\min_{\theta'} \underbrace{\mathbb{E}_{c_e \in \mathbf{E}} \left[ \left\| \epsilon_{\theta'}(\tau(c_e)) - \epsilon_\theta(\tau(c_t)) \right\|_2^2 \right]}_{L_1} + \underbrace{\left\| \epsilon_{\theta'}(\tau(c_n)) - \epsilon_\theta(\tau(c_n)) \right\|_2^2}_{L_2} \tag{2}$$

While this naive principle is simple and effective in erasing the specific concept, choosing a fixed neutral concept as the target concept for all concepts to be erased is obviously not optimal (i.e., $c_t = c_n \ \forall c_e \in \mathbf{E}$). Intuitively, remapping the visual concept "English Springer" to "Dog" will likely cause a less drastic change on the model's parameter compared to remapping it to a neutral concept such as "A photo" or " ", leading to a better preservation performance on other concepts. In the following section, we will provide a series of evidence to support this intuition which leads us to a principled approach on choosing an optimal target concept for each concept to be erased.

## 3 QUANTIFYING IMPACT OF ERASING CONCEPTS

**Netfive Dataset.** Two main challenges in evaluating an erasing method are: (1) How to verify whether a concept is present in the generated images or not? (2) How to ensure the evaluation is diverse enough to cover the output space of the model which is of infinite possibilities? To tackle these two challenges, we propose an evaluation dataset called NetFive, which consists of 25 concepts from the ImageNet dataset, for which we can leverage the pre-trained classification model to verify the presence of the concepts in the generated images. We also ensure the diversity of the evaluation by generating 500 samples for each concept. More specifically, we choose a total of five subsets of concepts, each subset contains one anchor concept, e.g., "English Springer" and four related concepts, ranked by their closeness to the anchor concept as follows: (the details and rationale of choosing the concepts are provided in Appendix C.1)

- **Dog:** English springer, Clumber spaniel, English setter, Blenheim spaniel, Border collie.
- **Vehicle:** Garbage truck, Moving van, Fire engine, Ambulance, School bus.
- **Instrument:** French horn, Basson, Trombone, Oboe, Saxophone.
- **Building:** Church, Monastery, Bell cote, Dome, Library.
- **Equipment:** Cassette Player, Polaroid camera, Loudspeaker, Typewriter keyboard, Projector.

**Metric to measure the generation capability of the model.** Because of intentionally choosing all concepts from the ImageNet dataset, we can leverage the pre-trained classification model to detect the presence of the concept in the generated image, More specifically, given a model $G_\theta$, we generate $k = 500$ images with the input description $c$, and then measure using the metrics:

- **Detection Score (DS-1/DS-5):** # of samples that are classified as the concept $c$ in top-1 or top-5 predictions / $k$ (i.e., top-k accuracy).
- **Confident Score (CS-1/CS-5):** Average confident score w.r.t. the concept $c$ in top-1 or top-5 predictions, otherwise, the confident score is set to be 0. This metric indicates how good the model $G_\theta$ is when generating the concept $c$. A higher score means that the concept $c$ is more likely appeared in the generated images.

For all metrics, higher values indicate better generation capability of the model on concept $c$. In the rest of the paper, we use the Detection Score (DS-5) as the main metric, while results for other metrics are provided in Appendix C.2.

We present $G_0(c_j)$ and $G_{c_e}(c_j)$ as the generation capability of the original model and the sanitized model on the same query concept $c_j$ after erasing concept $c_e$, respectively. With this, we can measure the impact of erasing concept $c_e$ on the generation capability of concept $c_j$ by computing the difference between $G_0(c_j)$ and $G_{c_e}(c_j)$, i.e., $\Delta(c_e, c_j) = G_0(c_j) - G_{c_e}(c_j)$.

### 3.1 TARGETING TO A GENERIC CONCEPT

We first analyze the impact of choosing a generic concept such as an empty " " as the target concept for erasure. The top subfigure in Figure 1 shows **a sample analysis** of the impact of erasing concept "English Springer" to the generation capability of all NetFive concepts. Each column corresponds to one concept $c_j$ in the NetFive dataset. The blue bar represents the generation capability $G_{c_e}(c_j)$ of the sanitized model on concept $c_j$, while the total height of the stack is the generation capability $G_0(c_j)$ of the original model on concept $c_j$. The red bar, therefore, represents the gap of generation capability between the sanitized model and the original model, i.e., $\Delta(c_e, c_j) = G_0(c_j) - G_{c_e}(c_j)$,

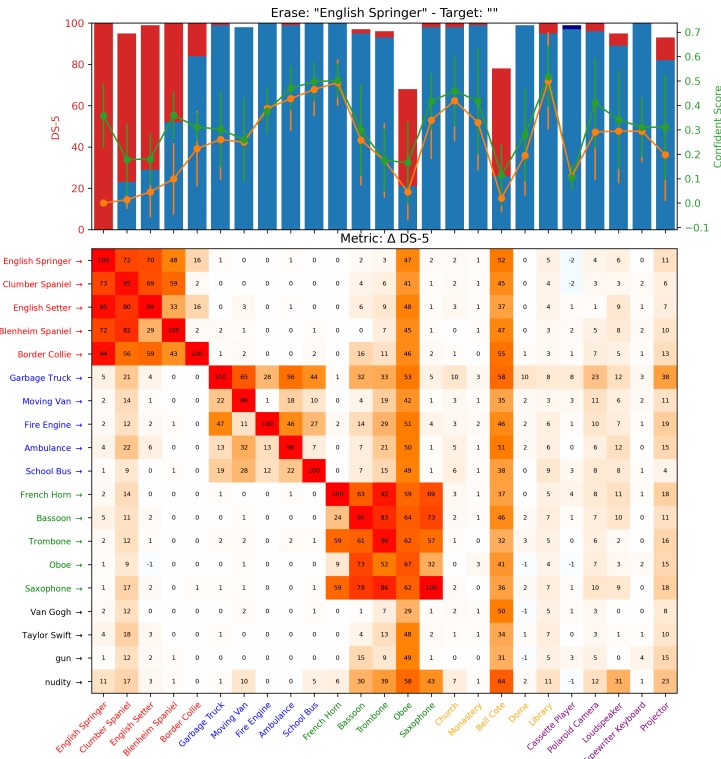

Figure 1: Analysis of the impact of choosing empty concept as the target concept for erasure. Complete details are provided in Figure 9.

The higher the total height of the stack, the higher the generation capability of the original model on concept $c_j$, while the higher the red bar, the higher the difference between the generation capability of the sanitized model and the original model. We also provide the confidence score of the classifier when predicting the concept from the generated images from the original model (the green line) and the sanitized model (the orange line).

**Sample Analysis.** It can be seen from the top subfigure in Figure 1 that the concept $c_e$ "English Springer" has been successfully removed from the sanitized model as evidenced by $G_{c_e}$("English Springer") = 0. It can also be seen that all closely related concepts to concept $c_e$ "English Springer" are affected by the erasure of this concept, evidenced by the significant drop in the generation capability $G_{c_e}(c_j) < G_0(c_j)$, as well as the corresponding confidence scores. In contrast, the other concepts that are not closely related to the concept $c_e$ are less affected, evidenced by the unchanged generation capability $G_{c_e}(c_j) \approx G_0(c_j)$. Interestingly, the two abnormal concepts that are not related to the to-be-erased concept $c_e$ but are also affected, "Bell Cote" and "Oboe". These two concepts have low generation capability even before the erasure of concept $c_e$, i.e., $G_0(c_j) \approx 60\%$, compared to other concepts which have $G_0(c_j) \approx 100\%$.

**Systematic Analysis.** Figure 1 shows the impact of erasing one concept to all the other concepts in the NetFive dataset, where each row corresponds to erasing one concept, and the first row corresponds to erasing the concept "English Springer" as shown in the top subfigure. Each cell in the matrix represents the drop of the generation capability of concept $c_j$ after erasing concept $c_e$, i.e., $\Delta(c_e, c_j) = G_0(c_j) - G_{c_e}(c_j)$. A deeper red color indicates a larger drop of the generation capability of concept $c_j$ after erasing concept $c_e$. The matrix can be interpreted as a concept graph, showing the impact of one concept on other concepts.

There are several intriguing observations that can be made from Figure 1:

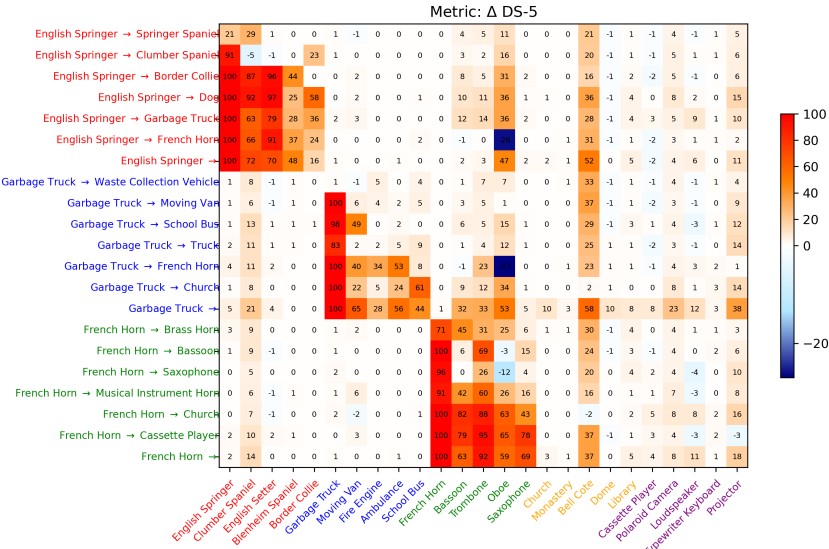

Figure 2: Analysis of the impact of choosing a specific concept as the target concept for erasure. Full details are provided in Figure 12.

- **Locality:** The concept graph is sparse and localized, which means that the impact of erasing one concept does not **strongly** spread to all the other concepts but only local concepts that are semantically related to the erased concept $c_e$.

- **Asymmetry:** The concept graph is asymmetric such that the impact of erasing concept $c_e$ on concept $c_j$ is not the same as the impact of erasing concept $c_j$ on concept $c_e$.

- **Abnormal:** The two abnormal concepts "Bell Cote" and "Oboe", which have low generation capability to begin with, are sensitive to the erasure of any concept.

In the case of erasing exclusive concepts, such as "Taylor Swift", "Van Gogh", "gun", and "nudity" as shown in the last four rows of the figure, the impact of erasing these concepts to all NetFive concepts are also limited, except for the two abnormal concepts, which supports the above observations on the concept graph.

**Results with Stable Diffusion version 2.1** Figure 13 shows the impact of choosing the empty concept as the target concept for erasure with Stable Diffusion v2.1 It can be seen that our earlier observations are still valid, except that the abnormal concepts which have low generation capability and are sensitive to the erasure of any concept are now "Bell Cote" and "Projector". This consistency indicates the generalization of the observations on the concept graph of different models, trained on different datasets and settings.

## 3.2 TARGETING TO A SPECIFIC CONCEPT

We now compare different strategies of choosing the target concept for erasure to see how the choice of the target concept affects the preservation of other concepts. In each subset, we choose to erase a same concept $c_e$ but with seven different target concepts $c_t$, in the following order: ($1^{st}$) a synonym of $c_e$, ($2^{nd}, 3^{rd}$) two semantically related concepts but not synonyms, ($4^{th}$) a general, upper-level concept that covers the subset, (e.g. "Dog" for the concept "English Springer") ($5^{th}, 6^{th}$) two semantically unrelated concepts, and ($7^{th}$) an empty concept. The explanation of finding a synonym to an anchor concept is provided in Appendix C.3.

It can be seen from Figure 2 that the preservation performance highly depends on the choice of the target concept. More specifically, there are several notable observations:

- **Locality:** Regardless of the choice of the target concept, the impact of erasing one concept is still sparse and localized.

- **Abnormal:** The two abnormal concepts "Bell Cote" and "Oboe" are still sensitive to the erasure of any concept regardless of the choice of the target concept.

- **Synonym ✗✗✗:** Mapping to a synonym of the anchor concept leads to a minimal change as evidenced by the lowest $\Delta(c_e, c_j)$ for all $c_j$. However, it also the least effective in erasing the undesirable concept $c_e$.

- **All Unrelated Concepts ✗✗:** Mapping to semantically unrelated concepts demonstrates the similar performance as the generic concept " ", as evidenced by the similar $\Delta(c_e, c_j)$ between the three last rows ($5^{th} - 7^{th}$) in each subset.

- **General concepts ✗:** While choosing a general concept as the target concept is intuitive and reasonable, it does not necessary lead to good preservation performance. For example, "English Springer" $\rightarrow$ "Dog" or "French Horn" $\rightarrow$ "Musical Instrument Horn" still cause a drop on related concepts in their respective subsets. Moreover, there are also small drops on erasing performance compared to other strategies, shown by DS-5 of 83%, 91%, and 78% when mapping "Garbage Truck" to "Truck", "French Horn" to "Musical Instrument Horn", and "Cassette Player" to "Audio Device", respectively. The two observations indicate that choosing a general concept is not an optimal strategy.

- **In-class ✓:** The highest preservation performance which is consistently observed in all five subsets is achieved when the target concept is a closely related concept to $c_e$. For example, "English Springer" $\rightarrow$ "Clumber spaniel" or "Garbage Truck" $\rightarrow$ "Moving Van" or "School Bus" in the "Dog" and "Vehicle" subsets.

## 4 PROPOSED METHOD: ADAPTIVE GUIDED ERASURE

Motivated by the observations in Section 3, we propose to select the target concept for erasure adaptively for each query concept to mitigate the side effects of erasing undesirable concepts in diffusion models. More specifically, the ideal target concept should satisfy the following properties:

- It should not be a synonym of the query concept that resembles a similar visual appearance (e.g., "nudity" to "naked" or "nude", or "Garbage Truck" to "Waste Collection Vehicle"). This ensures that the erasure performance on the query concept remains effective.

- It should be closely related to the query concept but not identical (e.g., "English Springer" to "Clumber Spaniel", or "Garbage Truck" to "Moving Van"). This helps preserve the model's generation capabilities on other concepts. As suggested by the locality property of the concept graph, changes in the model's output can be used to identify these locally related concepts.

Although it is possible to manually select the ideal target concept for each query concept based on the above properties, this approach is not scalable for a large erasing set $\mathbf{E}$. Therefore, we propose an optimization-based approach to automatically and adaptively find the optimal target concept for each query concept. Specifically, we aim to solve the following optimization problem:

$$\min_{\theta'} \mathbb{E}_{c_e \in \mathbf{E}} \max_{c_t \in \mathcal{C}} \left[ \underbrace{\|\epsilon_{\theta'}(\tau(c_e)) - \epsilon_\theta(\tau(c_t))\|_2^2}_{L_1} + \lambda \underbrace{\|\epsilon_{\theta'}(\tau(c_t)) - \epsilon_\theta(\tau(c_t))\|_2^2}_{L_2} \right] \tag{3}$$

where $\lambda$ is a trade-off hyperparameter and $\mathcal{C}$ is the search space of target concepts $c_t$.

**Minimizing the objective** $L_1$ w.r.t. $\theta'$ ensures that the output of the sanitized model for the query concept $c_e$ is close to the output of the original model but for the target concept $c_t$, which serves the purpose of erasing the undesirable concept $c_e$. Meanwhile, **minimizing the objective** $L_2$ w.r.t. $\theta'$ ensures that the output of the two models remain similar for the same input concept $c_t$, preserving the model's capability on the remaining concepts. In summary, the **outer minimization problem** optimizes the sanitized model parameters $\theta'$ to simultaneously erase undesirable concepts and preserve the model's functionality for other concepts.

On the other hand, **maximizing the objective** $L_1$ w.r.t. $c_t$ ensures that the solution $c_t^*$ is not a synonym of the query concept $c_e$, while **maximizing the objective** $L_2$ w.r.t. $c_t$ finds a sensitive

concept to the change of the model's parameter $\theta \to \theta'$. This helps identify the most related concept to the query concept $c_e$, as suggested by the locality property of the concept graph in Section 3. By maximizing both $L_1$ and $L_2$ w.r.t. $c_t$, we ensure that the solution $c_t^*$ satisfies both key properties of the ideal target concept. We provide empirical evidence in Section 5.2 and Appendix D.6, showing that the intermediate value of the solution $c_t^*$ from the optimization problem equation 3 aligns with the above analysis.

**Optimization Details.** Since the concept space $\mathcal{C}$ is discrete and finite, the straightforward approach is to enumerate all the concepts in $\mathcal{C}$ and select the one that maximizes the total loss in each optimization step of the outer minimization problem. However, this approach is computationally prohibitive for large $\mathcal{C}$. Moreover, some concepts can be complex and may not be interpreted as a single concept in the concept space $\mathcal{C}$. To address this, we formulate the target concept as a combination of multiple concepts in the concept space $\mathcal{C}$, i.e., $\tau(c_t) = \mathbf{G}(\pi) \odot T_{\mathcal{C}}$, where $\mathbf{G}$ is the Gumbel-Softmax operator, $\pi \in \mathcal{R}^{|\mathcal{C}|}$ is a learnable variable, and $T_{\mathcal{C}}$ is the textual embedding matrix of the entire concept space $\mathcal{C}$. We choose the Gumbel-Softmax operator with a temperature less than 1 to ensure that the target concept is a combination of a few main concepts rather than a mixture of all concepts in $\mathcal{C}$. The optimization problem equation 3 can be rewritten as follows:

$$\min_{\theta'} \mathbb{E}_{c_e \in \mathbf{E}} \max_{\pi} \left[ \underbrace{\|\epsilon_{\theta'}(\tau(c_e)) - \epsilon_{\theta}(\mathbf{G}(\pi) \odot T_{\mathcal{C}})\|_2^2}_{L_1} + \lambda \underbrace{\|\epsilon_{\theta'}(\mathbf{G}(\pi) \odot T_{\mathcal{C}}) - \epsilon_{\theta}(\mathbf{G}(\pi) \odot T_{\mathcal{C}})\|_2^2}_{L_2} \right]$$
(4)

in which the objective of the inner-max is transformed to find the continuous weight $\pi$ instead of the discrete concept $c_t$. This trick allows us to increase the richness and expressiveness of the target concept $c_t$ as well as the optimization efficiency. We provide further details in Appendix B.

## 5 EXPERIMENTS

In this section, we demonstrate the effectiveness of our method in erasing various concepts from the foundation model, including object-related concepts, artistic styles, and NSFW attributes. We compare our approach with the state-of-the-art erasure methods including AP (Bui et al., 2024b), ESD (Gandikota et al., 2023), UCE (Gandikota et al., 2024), CA (Kumari et al., 2023), and MACE (Lu et al., 2024). Our experiments follow the same setup as in Bui et al. (2024b); Gandikota et al. (2023; 2024). Specifically, we use Stable Diffusion (SD) version 1.4 as the foundation model, and fine-tune the model for 1000 steps with a batch size of 1, using the Adam optimizer with a learning rate of $\alpha = 10^{-5}$.

Further implementation details and analyses are provided in the appendix, including qualitative results (Section D.8), an examination of the impact of vocabularies (Section D.2) and hyperparameters (Section D.3), as well as an analysis of the search for the optimal target concepts (Section D.6). We strongly recommend referring to the appendix for a deeper understanding of our method and experiments.

### 5.1 ERASING OBJECT-RELATED CONCEPTS

**Setting.** In this experiment, we assess our method's ability to erase object-related concepts, such as "Dog" or "Cat", from a foundational model. We utilize the Imagenette dataset [1], a subset of ImageNet (Deng et al., 2009), which contains 10 easily recognizable classes as suggested in (Gandikota et al., 2023). We conduct four different erasing tasks, each involving the simultaneous erasure of five classes while preserving the other five, generating 500 images per class.

**Metrics.** **Erasing performance** is measured using the *Erasing Success Rate (ESR-k)*, which calculates the percentage of generated images where the "to-be-erased" classes are not detected in the top-k predictions. **Preserving performance** is evaluated using the *Preserving Success Rate (PSR-k)*

---

[1] https://github.com/fastai/imagenette

to measure how well "to-be-preserved" classes are detected, along with *FID* and *CLIP* scores on the COCO 30K validation set to assess preservation of common concepts. For example, in Table 1, the average PSR-5 of the original SD model is 97.6%, which means that 97.6% of the generated images contain the object-related concepts in the top-5 predictions, indicating the strong ability to generate object-related concepts accurately of the foundation model.

**Results.** In term of erasure, all baselines perform well, with the lowest ESR-1 and ESR-5 scores being 95.5% and 88.9% respectively, indicating that only a small proportion of the generated images retain the object-related concepts. The UCE method achieves a perfect 100% ESR-1 and ESR-5, while our method reaches 98.1% ESR-1 and 95.7% ESR-5, slightly below AP, CA, and MACE.

However, when it comes to concept preservation, the baselines, particularly UCE, perform poorly, with the most recent SOTA baseline AP showing the best PSR-1 and PSR-5 scores at 55.2% and 79.9%, respectively. In contrast, our method significantly outperforms the baselines, achieving 73.6% PSR-1 and 95.6% PSR-5, far exceeding AP and approaching the original SD model's performance. Our method also achieves the best FID score of 16.1 and CLIP score of 26.0, surpassing the second-best AP method by 0.2 and 0.1 points, respectively. These results demonstrate that our method not only effectively erases unwanted concepts but also excels in preserving other important concepts, closely matching the performance of the foundation model.

Table 1: Erasing object-related concepts. The best erasing results are highlighted in **bold**, and the second-best erasing results are highlighted in underline.

| Method | ESR-1↑ | ESR-5↑ | PSR-1↑ | PSR-5↑ | FID↓ | CLIP↑ |
|--------|--------|--------|--------|--------|------|-------|
| SD | $22.0 \pm 11.6$ | $2.4 \pm 1.4$ | $78.0 \pm 11.6$ | $97.6 \pm 1.4$ | 16.1 | 26.4 |
| ESD | $95.5 \pm 0.8$ | $88.9 \pm 1.0$ | $41.2 \pm 12.9$ | $56.1 \pm 12.4$ | 17.9 | 24.5 |
| UCE | $\mathbf{100 \pm 0.0}$ | $\mathbf{100 \pm 0.0}$ | $23.4 \pm 3.6$ | $49.5 \pm 8.0$ | 19.1 | 21.4 |
| CA | $98.4 \pm 0.3$ | $96.8 \pm 6.1$ | $44.2 \pm 9.7$ | $66.5 \pm 6.1$ | 16.6 | 25.8 |
| MACE | $\underline{99.3 \pm 0.3}$ | $\underline{97.6 \pm 1.2}$ | $47.4 \pm 12.0$ | $72.8 \pm 10.5$ | 16.9 | 24.9 |
| AP | $98.6 \pm 1.1$ | $96.1 \pm 2.7$ | $\underline{55.2 \pm 10.0}$ | $\underline{79.9 \pm 2.8}$ | $\underline{16.3}$ | $\underline{26.1}$ |
| Ours | $98.1 \pm 1.1$ | $95.7 \pm 2.5$ | $\mathbf{73.6 \pm 9.8}$ | $\mathbf{95.6 \pm 1.1}$ | **16.1** | **26.0** |

## 5.2 ERASING NSFW CONCEPTS

**Setting.** In this experiment, we focus on unlearning Not-Safe-For-Work (NSFW) attributes like "nudity" from the model's capability. We follow the same setting as in (Gandikota et al., 2023), focusing exclusively on fine-tuning the non-cross-attention modules. To generate NSFW images, we employ I2P prompts (Schramowski et al., 2023b) and generate a dataset comprising 4703 images with attributes encompassing sexual, violent, and racist content.

**Metrics.** We utilize the detector (Praneet, 2019) which can accurately detect several types of exposed body parts to recognize the presence of the nudity concept in the generated images. The detector (Praneet, 2019) provides multi-label predictions with associated confidence scores, allowing us to adjust the threshold and control the trade-off between the num-

Table 2: Evaluation on the nudity erasure setting.

| | NER-0.3↓ | NER-0.5↓ | NER-0.7↓ | NER-0.8↓ | FID↓ |
|-----|----------|----------|----------|----------|-------|
| CA | 13.84 | 9.27 | 4.74 | 1.68 | 20.76 |
| UCE | 6.87 | 3.42 | 0.68 | 0.21 | 15.98 |
| ESD | 5.32 | 2.36 | 0.74 | 0.23 | 17.14 |
| AP | 3.64 | 1.70 | 0.40 | 0.06 | 15.52 |
| Ours | 5.06 | 1.53 | 0.32 | 0.04 | 14.20 |

ber of detected body parts and the confidence of the detection, i.e., the higher the threshold, the fewer the number of detected body parts. **Erasing performance** is measured using the *Nudity Exposure Rate (NER-k)*, which measures the ratio of images with any exposed body parts detected by the detector (Praneet, 2019) over the total number of generated images with a confidence score greater than the threshold $k$. For example, in Table 2, with the threshold set at 0.5, the NER score for the CA model stands at 9.27%, indicating that 9.27% of the generated images contain signs of nudity concept from the detector's perspective. **Preserving performance** is evaluated using FID score on COCO 30K validation set.

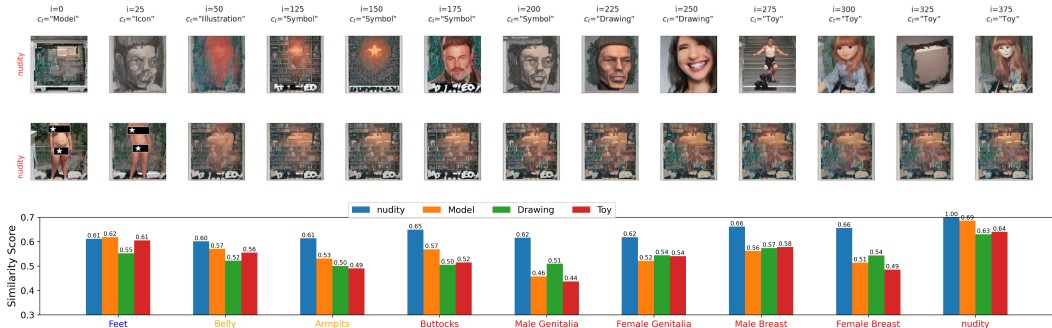

Figure 4: Top/a: Intermediate results of the search process, with images generated from the most sensitive concepts $c_t$ found by our method and $c_e$ at the same optimization step. Bottom/b: Similarity between nudity attributes and keywords.

**Results.** Table 2 shows the NER score across thresholds ranging from 0.3 to 0.8. With $k = 0.5$, CA, ESD, UCE, and AP achieve 9.27%, 2.36%, 3.42%, and 1.70% NER, respectively, while our method achieves 1.53% NER, the lowest among the baselines, indicating the highest erasing performance. This result remains consistent across different thresholds, emphasizing the robustness of our method in erasing NSFW content. In term of preserving performance, our method achieves the best FID score of 14.20, a significant improvement over the second and third-best AP and UCE methods at 15.52 and 15.98, respectively, indicating that our method can simultaneously erase a concept while preserving other concepts effectively.

Figure 3 provides detailed statistics of different exposed body parts in the generated images. It can be seen that in the original SD model, among all the body parts, the female breast is the most detected body part in the generated images, accounting for more than 320 images out of the total 4703 images. Both baselines, ESD and UCE, as well as our method, achieve a significant reduction in the number of detected body parts, with our method achieving the lowest number among the baselines. Our method also achieves the lowest number of detected body parts for the most sensitive body parts, only surpassing the baselines for feet which is a less sensitive body part.

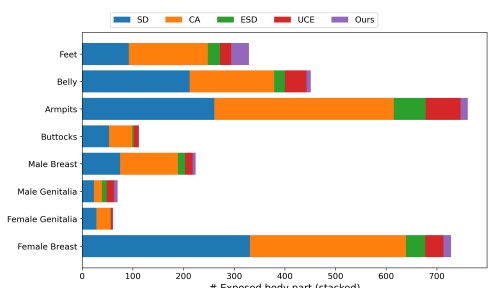

Figure 3: Number of exposed body parts counted in all generated images with threshold 0.5.

**Searching for the optimal target concepts.** We investigate the search for optimal target concepts $c_t$ by visualizing the generated images from $c_t$ found by our method and the corresponding output of $c_e$ at the same optimization step as shown in Figure 4a. Additionally, we present the similarity scores between the nudity attributes detected by the detector and $c_t$ as in Figure 4b. Firstly, the to-be-erased concept $c_e$ ("nudity") is closely related to the sensitive body parts such as "breasts" and "genitalia", explaining why removing the "nudity" concept also eliminates these sensitive attributes. On the other hand, as discussed in Section 4, the target concept $c_t$ is designed to be locally aligned with $c_e$ but not exactly the same as $c_e$. In our results, all intermediate concepts $c_t$ such as "Model", "Drawing", and "Toy" are highly correlated with the "nudity" concept satisifying the first condition. However, while being highly correlated with less sensitive parts such as "Feet", they are less correlated with more sensitive ones like "breasts" and "genitalia", meeting the second condition. It is a worth recall that in Equation 3, $c_t$ serves as a retained concept to preserve the model's capabilities. Therefore, the strong correlation between $c_t$ and "Feet", and its weaker connection with other sensitive parts, explains the interesting advantage of our method that it can still retain the "Feet" concept while successfully erasing others, as observed in Figure 3. We provide more results and analyses in Appendix D.6.

## 5.3 ERASING ARTISTIC CONCEPTS

**Setting.** In this experiment, we evaluate our method's ability to erase artistic style concepts. We focus on five well-known artists with highly recognizable styles that are commonly mimicked by text-to-image generative models, including "Kelly Mckernan", "Thomas Kinkade", "Tyler Edlin", "Kilian Eng", and "Ajin Demi Human" as in (Gandikota et al., 2023). The experiment involves five tasks, each aiming to erase one artist's style while preserving the others.

**Metrics.** A major challenge in this setting is the lack of a reliable detector for identifying the presence of artistic styles in generated images. Human-evaluation is avoided due to the high cost, time-consuming, not scalable, and more importantly, easily biased. To overcome this, we utilize the CLIP score [2] to measure the alignment between the generated images and the textual prompts, which has been shown to be effective in similar tasks (Gandikota et al., 2023). To enhance the correctness, we make use of a list of long textual prompts that are designed exclusively for each artist (credits to (Gandikota et al., 2023)), combined with 5 seeds per prompt to generate 200 images for each artist across all methods. This approach allows the use of the CLIP score as a more meaningful measurement to evaluate the erasing and preserving performance. We also use LPIPS (Zhang et al., 2018) to measure the distortion in generated images by the original SD model and editing methods, where a low LPIPS score indicates less distortion between two sets of images. However, as LPIPS is designed for quantitative comparison between two outputs, it might not be as good as CLIP score in order to verify the presence of a specific concept in the generated images, and should be used as a complementary metric to CLIP score.

**Results.** Figure 5 presents the results of the artistic style erasure task. In term of CLIP score, it can be seen that our method along with AP and UCE lie in the Pareto frontier (with the ideal point being the top-left corner), with a clear trade-off between erasing and preserving performance. For example, AP achieves better erasing performance than our method with a CLIP score of 21.57 and 22.44, respectively, but our method outperforms AP in preserving performance with a CLIP score of 30.45 and 30.13, respectively. However, in term of LPIPS score, our method clearly achieves the best trade-off between erasing and preserving performance, with the erasing score of 0.80 and preserving score of 0.44, while AP achieves the erasing score of 0.78 and preserving score of 0.47.

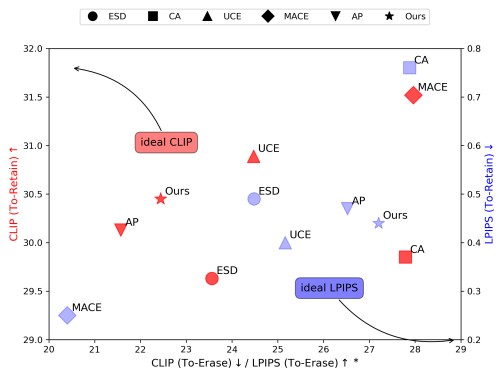

Figure 5: CLIP and LPIPS scores of the artistic style erasure task. (*) LPIPS at x-axis is scaled by 34 for better visualization. Full results are in Table 6.

## 6 CONCLUSION

In this paper, we introduced the Adaptive Guided Erasure (AGE) method, a novel approach for concept erasure that addresses the limitations of existing fixed-target methods. By modeling the concept space as a graph and analyzing its geometric properties, we demonstrated that selecting a locally related target concept can minimize unintended side effects. AGE adapts target selection through a minimax optimization, further enriched by representing targets as mixtures of single concepts. Our experiments show that AGE outperforms state-of-the-art methods, effectively erasing undesirable concepts while preserving benign ones across various tasks. Besides the method, we also provided the first comprehensive study of the concept space structure, providing new intriguing insights that shed light on the concept space geometry. We believe that these insights will inspire future research on understanding and manipulating the concept space for various applications.

---

[2]https://lightning.ai/docs/torchmetrics/stable/multimodal/clip_score.html

ACKNOWLEDGEMENTS

The Commonwealth of Australia (represented by the Advanced Strategic Capabilities Accelerator) supports this research through a Defence Science Partnerships agreement. Dinh Phung further acknowledges the support from the Australian Research Council (ARC) Discovery Project DP250100262. The authors would like to thank the anonymous reviewers for their insightful feedback and valuable suggestions, which have significantly enhanced the quality of this work.

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

# Appendix

## Table of Contents

## A    RELATED WORK

Existing methods for removing or unlearning undesirable concepts from text-to-image generative models can be categorized into four main categories: (1) pre-processing dataset filtering, (2) post-processing screening, (3) model fine-tuning, and (4) in-generation guidance. Each approach has its advantages and drawbacks. While pre-processing and post-processing methods are effective, they may not address the underlying tendencies of models to generate inappropriate content (SmithMano, 2022). Fine-tuning approaches provide a more direct solution but often struggle with maintaining model performance while removing specific concepts. In-generation guidance offers flexibility but can be resource-intensive and requires careful tuning.

**Pre-processing dataset filtering**    is the most straightforward method, involving mechanisms to detect and remove or flag undesirable content in the training data. This generally requires a pre-trained detector to identify objectionable content within both the original dataset and any newly added data. However, the primary drawback of this approach is that it necessitates retraining the model from scratch, which is computationally expensive and impractical for dynamic erasure requests. Examples of this include Stable Diffusion v2.0 (StabilityAI, 2022), which uses an NSFW detector to filter the LAION-5B dataset (Schuhmann et al., 2022), and Adobe Firefly, which is trained on licensed and public-domain content to ensure commercial safety (Rao, 2023). DALL·E 3 (Shi et al., 2020) further refines this by subdividing the NSFW concept into specific categories and applying individualized detectors. Despite significant efforts in curating training data, pre-processing methods often leave models inadequately sanitized, as shown in Gandikota et al. (2023).

**Post-processing screening**    methods identify inappropriate content in generated images after inference. Images flagged by detectors are either blurred or blocked before being shown to users. This approach is employed by organizations such as OpenAI (DALL-E), StabilityAI (Stable Diffusion), and Midjourney Inc. DALL·E 3, for instance, enhances this approach by training standalone detectors for various concepts such as race and gender. While post-processing is highly effective, it can be vulnerable to adversarial attacks. As demonstrated in Yang et al. (2024), adversaries can use techniques like Boundary Attack (Brendel et al., 2017) to circumvent such filters.

**In-generation guidance** methods intervene directly during the image generation process, often using techniques such as handcrafted textual blacklisting (Shi et al., 2020) or employing large language models for prompt engineering and safety classification. Safe Latent Diffusion (SLD) (Schramowski et al., 2023a) takes an alternative approach by leveraging inappropriate knowledge encoded in pre-trained models for reverse guidance during generation.

**Model parameter fine-tuning** offers a more scalable solution by fine-tuning pre-trained foundation models to remove unwanted concepts without retraining from scratch. This approach modifies the model's parameters, making it unnecessary to use detectors in post-processing. Consequently, releasing a sanitized model is as simple as providing a new checkpoint, which is less vulnerable to adversarial evasion. Fine-tuning methods have received considerable attention, with notable works including TIME (Orgad et al., 2023), ESD (Gandikota et al., 2023), Concept Ablation (Kumari et al., 2023), UCE (Gandikota et al., 2024), Forget-Me-Not (Zhang et al., 2023), and MACE (Lu et al., 2024). However, these models remain susceptible to inversion attacks (Pham et al., 2023), where adversaries can use textual inversion techniques (Gal et al., 2022) to learn and regenerate the removed concepts using the sanitized model.

Within fine-tuning, there are two main branches of concept erasing techniques: (1) Attention-based, and (2) Output-based or optimization-based.

*Attention-based* methods (Zhang et al., 2023; Orgad et al., 2023; Kumari et al., 2023; Gandikota et al., 2024; Lu et al., 2024) focus on modifying the attention mechanisms within models to remove undesirable concepts. In Latent Diffusion Models (LDMs), for instance, the textual conditions are embedded via a pre-trained CLIP model and injected into the cross-attention layers of the UNet model (Rombach et al., 2022; Ramesh et al., 2022). To remove an unwanted concept, the attention mechanism between the textual condition and visual information flow is altered. For example, in TIME (Orgad et al., 2023), the authors propose the following optimization:

$$\min_{W'} \sum_{c_e \in E} \|W' c_e - v_t^*\|_2^2 + \lambda \|W' - W\|_2^2 \tag{5}$$

where $W$ represents the original cross-attention weights, $W'$ the fine-tuned weights, $c_e$ the embedding of the unwanted concept, and $v_t^*$ the targeted vector. Setting $v_t^* = W c_t$, where $c_t$ is the embedding of a desired concept, allows the model to project the unwanted concept toward a more acceptable one. Follow-up works (Zhang et al., 2023; Gandikota et al., 2024; Lu et al., 2024) share this principle. Specifically, Forget-Me-Not (Zhang et al., 2023) introduces an attention resteering method that minimizes the L2 norm of the attention maps related to the unwanted concept. UCE (Gandikota et al., 2024) extends TIME by proposing a preservation term that allows the retention of certain concepts while erasing others. MACE (Lu et al., 2024) improves the generality and specificity of concept erasure by employing LoRA modules (Hu et al., 2021) for each individual concept, combining them with the closed-form solution from TIME (Orgad et al., 2023). This category has two main advantages: (1) the Tikhonov regularization form of the objective function 5 allows for a closed-form solution, as demonstrated in (Orgad et al., 2023), and (2) it operates solely on textual embeddings, making it faster than optimization-based methods.

*Output-based* methods (Gandikota et al., 2023; Wu et al., 2024) focus on optimizing the output image by minimizing the difference between the predicted noise $\epsilon_{\theta'}(z_t, t, c_e)$ and the target noise $\epsilon_\theta(z_t, t, c_t)$. Unlike attention-based methods, this approach requires intermediate images $z_t$ sampled at various time steps $t$ during the diffusion process. While this method is computationally more expensive, it generally yields superior erasure results by directly optimizing the image, ensuring the removal of unwanted concepts (Gandikota et al., 2023). Our proposed method belongs to this category, but our insights into optimal target concepts are generalizable across both categories.

A recent addition to the field, SPM (Lyu et al., 2024), introduces one-dimensional adapters that, when combined with pre-trained LDMs, prevent the generation of images containing unwanted concepts. SPM introduces a new diffusion process $\hat{\epsilon} = \epsilon(x_t, c_t \mid \theta, \mathcal{M}_{c_e})$, where $\mathcal{M}_{c_e}$ is an adapter model trained to remove the undesirable concept $c_e$. While these adapters can be shared and reused across different models, the original model $\theta$ remains unchanged, allowing malicious users to remove the adapter and generate harmful content. Thus, SPM is less robust and practical compared to the other approaches discussed.

**Adversarial Concepts.**  The definition of *adversarial concepts*- those most affected by changes in model parameters when a concept is erased- was first introduced in (Bui et al., 2024b). Building on empirical observations that erasing different concepts impacts the remaining ones in various ways and that preserving a neutral concept alone is insufficient to maintain the model's capabilities, the authors propose a method for preserving sensitive concepts while erasing undesirable ones. This approach is similar to the adversarial training principle (Goodfellow et al., 2014; Bui et al., 2022; 2021; 2020) which has been successfully applied to improve the robustness of models against adversarial attacks. Our work is inspired by the idea of adversarial concepts but with several key differences: ❶ Instead of focusing on the to-be-preserved concept, we investigate the impact of the target concept on the remaining concepts, offering new insights into the selection and importance of the target concept in the erasure problem. ❷ We propose an improved optimization method that more accurately identifies adversarial concepts, explicitly excluding those identical to the to-be-erased concept—an aspect not considered in (Bui et al., 2024b). ❸ For the first time, we introduce the concept of a *concept graph* and its properties, providing a deeper understanding of concept relationships in the latent space.

# B    Further Details on the Adaptive Guided Erasure Method

In this section, we provide more details on the proposed adaptive guided erasure method.

**Algorithm.**  We first recall the central optimization problem in Equation equation 3:

$$\min_{\theta'} \mathbb{E}_{c_e \in \mathbf{E}} \max_{\pi} \left[ \underbrace{\|\epsilon_{\theta'}(\tau(c_e)) - \epsilon_\theta(\mathbf{G}(\pi) \odot T_\mathcal{C})\|_2^2}_{L_1} + \lambda \underbrace{\|\epsilon_{\theta'}(\mathbf{G}(\pi) \odot T_\mathcal{C}) - \epsilon_\theta(\mathbf{G}(\pi) \odot T_\mathcal{C})\|_2^2}_{L_2} \right]$$

where $\tau(\cdot)$ is the textual embedding function that embeds a concept into a textual embedding, $\mathbf{G}(\cdot)$ is the Gumbel-Softmax operator, and $\odot$ denotes the element-wise product. $T_\mathcal{C}$ is the textual embedding matrix of the entire concept space $\mathcal{C}$, i.e., $T_\mathcal{C} = \{\tau(c_1), \tau(c_2), \cdots, \tau(c_n)\}$ that can be pre-computed by the fixed textual encoder $\tau(\cdot)$ in the foundation model.

The algorithm to solve the min-max optimization problem is provided in Algorithm 1.  More specifically, given the foundation model $\theta$, concept space $\mathcal{C}$, and the erasing set $\mathbf{E}$, we precompute the embedding matrix $T_\mathcal{C}$. Since we erase multiple concepts simultaneously, each concept $c_e$ has an associated optimal target concept $c_t$. Therefore, we maintain a dictionary $\mathbf{D}$ to store the weights $\pi$ of the optimal target concepts for all concepts in the erasing set $\mathbf{E}$ throughout the optimization process. Initially, the weights $\pi$ are uniformly distributed across all concepts in $\mathcal{C}$, i.e., $\mathbf{D}[c_e] = [1/|\mathcal{C}|, 1/|\mathcal{C}|, \cdots, 1/|\mathcal{C}|]$.

During each iteration, we first sample a concept $c_e$ from the erasing set $\mathbf{E}$. Then, we retrieve the previous target concept $\pi$ of $c_e$ from the dictionary $\mathbf{D}$. By doing so, we can ensure the learning process is stable and continuous from previous learning process. We then compute the gradient of the loss function w.r.t. the weight $\pi$ and update the weight $\pi$. It can be done iteratively for $N_{\text{iter}}$ times, however, in our experiment, we just simply set $N_{\text{iter}}$ to be 1 for simplicity. After having the updated weight $\pi$, we update the model parameters $\theta'$ by performing a gradient descent step, as well as the dictionary $\mathbf{D}$.

A detailed analysis on the impact of hyperparameters is provided in Section D.3. In all experiments, we simply set the trade-off $\lambda = 1.0$, the temperature $\gamma = 0.1$, learning rate $\eta = 0.001$, and $N_{\text{iter}} = 1$. Visualizations of the mixture of concepts can be found in Section D.5, and the process of searching for optimal target concepts is discussed in Section D.6.

**Limitation.**  A crucial aspect of our method is the concept space $\mathcal{C}$, which is used to search for the optimal target concept. As discussed earlier, we use the Gumbel-Softmax trick, which requires feedings the model with the embedding matrix $T_\mathcal{C}$ of all concepts in the concept space $\mathcal{C}$. However, this requires a large computational cost, especially when the concept space $\mathcal{C}$ is large. To mitigate the issue, we use a small set of concepts $\mathcal{C}_{c_e}$ which contains the most $k$ closest concepts to the

---

**Algorithm 1** Adaptive Guided Erasure Fine-tuning

---

**Input:** $\theta, T_{\mathcal{C}}, \mathbf{E}, \lambda, \gamma$. Searching hyperparameters: $\eta, N_{\text{iter}}$. Learning rate: $\alpha$
**Output:** $\theta^{'}$
$k \leftarrow 0, \theta^{'}_k \leftarrow \theta$
$\mathbf{D}[c_e] \leftarrow [1/|\mathcal{C}|, 1/|\mathcal{C}|, \cdots, 1/|\mathcal{C}|] \;\forall c_e \in \mathbf{E}$         ▷ Init the dictionary
**while** Not Converged **do**
    $c_e \sim \mathbf{E}$
    // Find the target concept $c_t$ w.r.t. current $c_e$
    $\pi \leftarrow \mathbf{D}[c_e]$         ▷ Retrieve the previous target concept
    **for** $i = 1$ to $N_{\text{iter}}$ **do**
        $\pi \leftarrow \pi + \eta\nabla_\pi \left[ L_1(\pi, \theta, \theta^{'}_k) + \lambda L_2(\pi, \theta, \theta^{'}_k) \right]$         ▷ Inner Max
    **end for**
    $\mathbf{D}[c_e] \leftarrow \pi$         ▷ Update the dictionary
    // Update model parameters
    $\theta^{'}_{k+1} \leftarrow \theta^{'}_k - \alpha\nabla_{\theta^{'}_k} \left[ L_1(\pi, \theta, \theta^{'}_k) + \lambda L_2(\pi, \theta, \theta^{'}_k) \right]$         ▷ Outer min
    $k \leftarrow k + 1$
**end while**

---

concept $c_e$ in the original concept space $\mathcal{C}$ for each concept $c_e$ to reduce the computational cost. We simply choose $k = 100$ for all experiments. The similarity between concepts is measured in textual embedding space $\tau(\cdot)$ by cosine similarity. We provide the analysis on the impact of choosing concept space $\mathcal{C}_{c_e}$ in Section D.2.

Additionally, as demonstrated in Section D.5, the mixture of concepts is not a simple linear combination of the concepts. Some transformations, like from English springer" to French Horn", are smooth, while others are not. This limits the potential of fully leveraging the concept mixture. We leave further exploration of smoother concept mixtures for future work.

## C  EXPERIMENTAL SETTINGS

### C.1  NETFIVE DATASET

Two main challenges in evaluating an erasing method are: (1) How to verify whether a concept is present in the generated images or not? (2) How to ensure the evaluation is diverse enough to cover the output space of the model which is of infinite possibilities? To tackle the two challenges, we propose a evaluation dataset called NetFive, which consists of 25 concepts from the ImageNet dataset, for which we can leverage the pre-trained classification model to verify the presence of the concepts in the generated images. We also ensure the diversity of the evaluation by generating 500 samples for each concept More specifically, we choose total five subsets of concepts, each subset contains one anchor concept, e.g., "English Springer" and four related concepts, ranked by their relative closeness to the anchor concept.

To choose the related concepts, we use the original SD model to generate 500 samples for each concept. We then use the pre-trained ResNet50 model to classify the generated samples. We then choose related concepts from top-10 most frequently appeared concepts in the output. In the end, we choose the following subsets of concepts:

- **Dog Subset:** English springer, Clumber spaniel (315), English setter (160), Blenheim spaniel (45), Border collie (39)

- **Vehicle Subset:** Garbage truck, Moving van (244), Fire engine (188), Ambulance (97), School bus (33)

- **Instrument Subset:** French horn, Basson (327), Trombone (210), Oboe (199), Saxophone (38)

- **Building Subset:** Church, Monastery (445), Bell cote (274), Dome (134), Library (17)

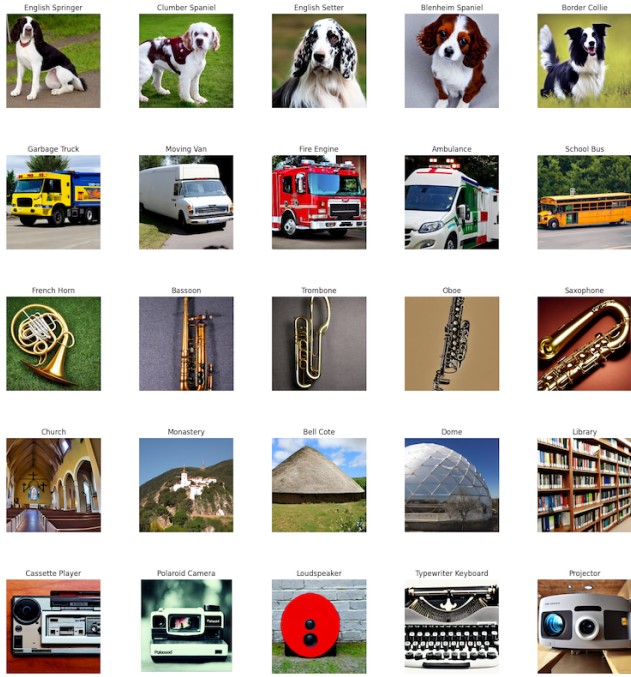

Figure 6: Sample images generated by the SD model from the NetFive dataset. Each row corresponds to one subset of concepts.

- **Equipment Subset:** Cassette Player, Polaroid camera (78), Loudspeaker (51), Typewriter keyboard (19), Projector (9)

Each subset contains one anchor concept, i.e., English springer, and four related concepts, ranked by the frequency of appearance of that concept in the top-10 predictions. For example, in 500 samples with the input description "English Springer", there are 315 samples that are classified as "Clumber spaniel" in the top-10 predictions. The larger the frequency, the more related the concept is to the anchor concept. We provide sample images generated by the SD model in Figure 6.

It is worth noting that, while most of the concepts in the dataset have high generation capability, i.e., the total height of the stack is almost 100% as shown in Figure 1, there are two concepts "Bell Cote" and "Oboe" that have low generation capability, with their total height of the stack around 60-80% as shown in Figure 1. Beside Figure 6, we also provide a failed sample from the "Oboe" concept in Figure 7 for reference.

## C.2 METRIC TO MEASURE THE GENERATION CAPABILITY OF THE MODEL

In image generative models, while common metrics such as FID and Inception Score are used to evaluate the quality of the generated images, there is no direct metric to evaluate the generation capability of the model on a specific concept. One of the main tasks in evaluating the generation capability is to detect the presence of the concept in the generated image. In our experiment, we intentionally choose all concepts from the ImageNet dataset, which allows us to leverage the pre-trained classification model to detect the presence of the concept in the generated image. Thus, we can indirectly evaluate the generation capability of the model on a specific concept.

More specifically, given a model $G_\theta$, we generate $k = 500$ images with the input description $c_j$, and then measure the two metrics:

- **Detection Score (DS-1/DS-5):** # of samples that are classified as the concept $c_j$ in top-1 or top-5 predictions / $k$. This metric indicates how many samples can be classified as the concept $c_j$ by the pre-trained classification model.

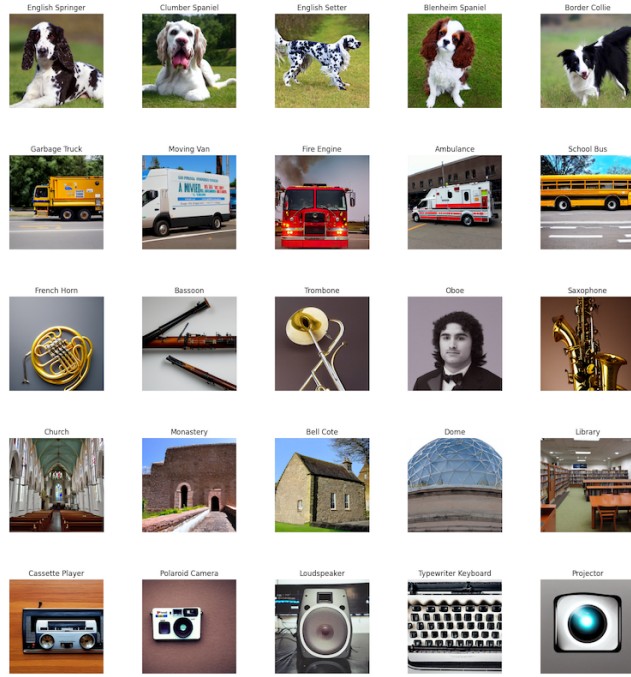

Figure 7: Sample images generated by the SD model from the NetFive dataset. Each row corresponds to one subset of concepts. **Failed sample from the "Oboe"** concept as showing a man instead of an oboe.

- **Confident Score (CS-1/CS-5):** Average confident score w.r.t. the concept $c_j$ in top-1 or top-5 predictions if the concept $c_j$ is detected in top-5 predictions, otherwise, the confident score is set to be 0. This metric indicates how confident the model is when generating the concept $c_j$. A higher score means that the concept $c_j$ is more likely to appear in the generated images.

It is worth noting that the confident score is designed to prefer a model that can generate more low-confident samples over a model that generates fewer high-confident samples. Additionally, the top-5 predictions are more reliable than the strict top-1 predictions, because the top-5 predictions can tolerate some errors in the model's prediction, e.g., a dog breed can be easily misclassified as another dog breed.

We present $G_0(c_j)$ and $G_{c_e}(c_j)$ as the generation capability of the original model and the sanitized model on the same query concept $c_j$ after erasing concept $c_e$, respectively. With this, we can measure the impact of erasing concept $c_e$ on the generation capability of concept $c_j$ by computing the difference between $G_0(c_j)$ and $G_{c_e}(c_j)$, i.e., $\Delta(c_e, c_j) = G_0(c_j) - G_{c_e}(c_j)$.

## C.3 FINDING A SYNONYM TO AN ANCHOR CONCEPT

In this section, we evaluate the erasing capability on the synonyms of object-related concepts, e.g., 'Church'. More specifically, we first utilize a set of tools including ChatGPT, Dictionary/Thesaurus.com, and Google image search to find the best synonyms for each target concept. To verify that these synonyms are indeed resembling the target concept, we then use the original model to generate images from the synonyms (e.g., 'a photo of Chapel'), and use the ResNet-50 model to classify the generated images. We then only keep the synonyms that have the top-5 accuracy higher than 50% to ensure that they are indeed generation-similar to the target concept. For some concepts we could not find any good synonyms, such as 'Golf ball' or 'Chain saw', except for some minor variations. We provide (top-1 and top-5) accuracy of the synonyms as below, as well as those numbers of target concepts, the higher the accuracy, the more similar the synonyms are to the target concept.

- **Cassette player (5.4;96.6)**: tape player (10.0;95.0), cassette deck (0.0;95.0), cassette recorder (2.5;92.5), tape deck (5.0;100.0)
- **Church (84.4;100.0)**: chapel (80.0;100.0), cathedral (50.0;100.0), minster (87.5;100.0), basilica (32.5;100.0)
- **Garbage truck (83.2;99.2)**: trash truck (87.5;97.5), refuse truck (80.0;100.0), waste collection vehicle (97.5;100.0), sanitation truck (47.5;100.0)
- **Parachute (95.2;99.2)**: skydiving chute (93.9;100.0), paraglider (100.0;100.0)
- **French horn (99.6;100.0)**: brass horn (32.5;95.0), double horn (37.5;65.0), German horn (22.5;80.0)
- **Chain saw (76.4;89.0)**: chainsaw (92.0;96.0), power saw (26.0;58.0)
- **Gas pump (69.0;97.0)**: fuel dispenser (50.0;97.5), petrol pump (85.0;100.0), fuel pump (47.5;65.0), gasoline pump (77.5;100.0), service station pump (47.5;87.5)
- **Tench (76.0;98.0)**: cyprinus tinca (60.0;95.0), cyprinus zeelt (52.5;100.0)
- **English springer (92.4;97.8)**: springer spaniel (95.0;100.0), springing spaniel (60.0;90.0), welsh springer spaniel (2.5;72.5), English cocker spaniel (32.5;77.5)
- **Golf ball (98.2;99.2)**: golfing ball (99.0;99.0)

## D  FURTHER ANALYSIS

### D.1  MEASURING THE IMPACT OF ERASING ON THE GENERATION CAPABILITY OF OTHER CONCEPTS

We would like to provide additional results with different metrics as follows while all results can be found in the project repository.

- Targeting to a generic concept, with DS-1 metric (Figure 8), DS-5 metric (Figure 9), CS-1 metric (Figure 10), CS-5 metric (Figure 11).
- Targeting to a specific concept with DS-5 metric (Figure 12).
- Targeting to a generic concept on Stable Diffusion version 2 with DS-5 metric (Figure 13).
- Targeting to a specific concept on Stable Diffusion version 2 with DS-5 metric (Figure 14).

**Observations based on DS-5 metric.**  To make the reading easier, we would like to recall the observations in Section 3 which are based on the DS-5 metric.

First, when mapping to a generic concept, we can observe the following properties of the concept graph:

- (Locality) The concept graph is sparse and localized, which means that the impact of erasing one concept does not **strongly** spread to all the other concepts but only a few concepts that are related to the erased concept $c_e$.
- (Asymmetry) The concept graph is asymmetric such that the impact of erasing concept $c_e$ on concept $c_j$ is not the same as the impact of erasing concept $c_j$ on concept $c_e$. Mathematically, $\Delta(c_i, c_j) \neq \Delta(c_j, c_i)$.
- (Abnormal) The two abnormal concepts "Bell Cote" and "Oboe", which have low generation capability to begin with, are sensitive to the erasure of any concept.

In the case of erasing exclusive concepts, such as "Taylor Swift", "Van Gogh", "gun", and "nudity" as shown in the last four rows of the figure, the impact of erasing these concepts to all NetFive concepts are also limited, except for the two abnormal concepts, which supports the above observations on the concept graph.

When mapping to a specific concept, we can observe the following properties that guide the choice of the target concept for erasure:

- (Locality) Regardless of the choice of the target concept, the impact of erasing one concept is still sparse and localized.

- (Abnormal) The two abnormal concepts "Bell Cote" and "Oboe" are still sensitive to the erasure of any concept regardless of the choice of the target concept.

- (Synonym ✗✗✗) Mapping to a synonym of the anchor concept leads to a minimal change as evidenced by the lowest $\Delta(c_e, c_j)$ for all $c_j$. However, it also the least effective in erasing the undesirable concept $c_e$.

- (All Unrelated Concepts ✗✗) Mapping to semantically unrelated concepts demonstrates the similar performance as the generic concept " ", as evidenced by the similar $\Delta(c_e, c_j)$ between the three last rows ($5^{th} - 7^{th}$) in each subset.

- (General concepts ✗) While choosing a general concept as the target concept is intuitive and reasonable, it does not necessary lead to good preservation performance. For example, "English Springer" → "Dog" or "French Horn" → "Musical Instrument Horn" still cause a drop on related concepts in their respective subsets. Moreover, there are also small drops on erasing performance compared to other strategies, shown by DS-5 of 83%, 91%, and 78% when mapping "Garbage Truck" to "Truck", "French Horn" to "Musical Instrument Horn", and "Cassette Player" to "Audio Device", respectively. The two observations indicate that choosing a general concept is not an optimal strategy.

- (In-class ✓) The highest preservation performance which is consistently observed in all five subsets is achieved when the target concept is a closely related concept to $c_e$. For example, "English Springer" → "Clumber spaniel" or "Garbage Truck" → "Moving Van" or "School Bus" in the "Dog" and "Vehicle" subsets.

**Results with Stable Diffusion version 2.** Figures 13 - 14 show the impact of choosing the empty concept and a specific concept as the target concept for erasure with Stable Diffusion version 2. It can be seen that the above observations are still valid, except for the abnormal concepts which have low generation capability and are sensitive to the erasure of any concept ("Bell Cote" and "Projector"). This consistency indicates the generalization of the observations on the concept graph of different models, trained on different datasets and settings.

In addition, we conduct four different experiments, each involving the simultaneous erasure of five classes from the Imagenette dataset while preserving the remaining five classes, generating 500 images per class. While we can successfully deploy the ESD method on Stable Diffusion v2.1, we are unable to do the same for the UCE method because of a implementation issue.

It can be seen from Table 3 that our method achieves significant improvements over the ESD method in both erasure and preservation performance, with a gain of 2.5% in ESR-5 and 8% in PSR-5. This indicates the generalizability of our method on different generative models.

| Method | ESR-1↑ | ESR-5↑ | PSR-1↑ | PSR-5↑ |
|--------|--------|--------|--------|--------|
| SD | 18.28 ± 8.97 | 1.80 ± 0.64 | 81.72 ± 8.97 | 98.20 ± 0.64 |
| ESD | 91.99 ± 6.35 | 87.83 ± 7.79 | 53.54 ± 8.22 | 75.45 ± 6.43 |
| AGE | 92.75 ± 6.96 | 90.27 ± 8.73 | 62.45 ± 13.30 | 83.43 ± 9.71 |

Table 3: Results of the experiment with Stable Diffusion version 2.

**Why Use the DS-5 Metric?** The CS-1 and CS-5 metrics measure the change in confidence scores associated with a query concept $c_j$ in the generated images, whereas the DS-1 and DS-5 metrics assess the change in detection status (presence or absence) of the query concept. Therefore, the CS-1 and CS-5 metrics are more sensitive to the change of the image content, and should be treated as a complementary analysis to the detection metrics.

Due to the high degree of class similarity in datasets like ImageNet, where, for instance, one dog breed may easily be misclassified as another, the DS-5 metric is more reliable than the DS-1 metric for verifying the presence of a concept in the generated images. The DS-5 metric's focus on detection across the top five predictions reduces the risk of misclassification, making it a more robust choice in such scenarios.

**Observations Based on Other Metrics** Figure 8 illustrates the impact of using the empty concept as the target for erasure, as measured by the DS-1 metric. While the results are not as pronounced as those seen with the DS-5 metric, similar patterns can be observed on the concept graph. For example, closely related concepts still show higher sensitivity to the erasure of a concept, as indicated by larger differences in the DS-1 metric (more red color or values exceeding 60%). The abnormal concepts such as "Oboe" remain sensitive to the erasure of any concept, with the smallest differences still exceeding 30%.

Similar trends are observed with the CS-1 metric, as shown in Figure 10, and with the CS-5 metric in Figure 11. However, to draw definitive conclusions, we emphasize that the impact of erasing a concept does not **strongly** propagate to all other concepts but tends to affect only local concepts that are semantically closely related to the erased concept $c_e$. Nonetheless, a weak impact on other concepts can still be detected using complementary metrics.

### D.2 IMPACT OF CHOOSING VOCABULARIES

**Object-related settings** Our method requires a concept space $\mathcal{C}$ predefined by users to search for the optimal target concept. In this experiment, we assess the impact of choosing different vocabularies for this task. The vocabularies compared include:

- ImageNet: A list containing all 1000 classes from the ImageNet dataset.
- Oxford: A list of the 3000 most common English words [3].
- CLIP: The CLIP token vocabulary, consisting of 49,408 tokens. While comprehensive, the CLIP vocabulary includes a significant number of nonsensical tokens, making it challenging to use.
- ChatGPT: A vocabulary generated by ChatGPT using the following prompt: "Provide k=100 objects that resemble a 'Cassette Player' but are not 'Cassette Player,' and export the list to a CSV file."

To reduce computational costs, we filter each original dictionary to a set of $k = 100$ concepts most similar to the target concept $c_e$, forming the concept space $\mathcal{C}_{c_e}$. Results indicate that the ChatGPT-generated vocabulary achieves the best preservation performance, with an impressive 96.49% top-5 accuracy—nearly matching the original model's performance. However, this vocabulary underperforms in the erasing task, as reflected by its lower ESR-1 and ESR-5 scores. Both the Oxford and CLIP vocabularies demonstrate similar erasing and preservation performance. Finally, the ImageNet vocabulary strikes the best balance between erasing and preserving, achieving the highest ESR-1 and ESR-5 scores, while maintaining respectable PSR-1 (80.84%) and PSR-5 (94.4%) scores, only slightly below the original model's performance. Consequently, we select the ImageNet vocabulary as the default vocabulary for all object-related experiments in the main paper. We provide the intermediate results of the erasing process for different vocabularies in Section D.6.

Table 4: Impact of choosing vocabularies on object-related settings.

| Vocab | ESR-1↑ | ESR-5↑ | PSR-1↑ | PSR-5↑ |
|---------|--------|--------|--------|--------|
| SD | 26.44 | 1.00 | 82.40 | 96.20 |
| ESD | 95.48 | 88.88 | 41.32 | 56.12 |
| UCE | 100.00 | 100.00 | 21.96 | 38.04 |
| ImageNet | 97.08 | 93.48 | 80.84 | 94.4 |
| Oxford | 93.48 | 87.68 | 66.88 | 85.40 |
| CLIP | 93.40 | 84.96 | 69.96 | 87.56 |
| ChatGPT | 83.60 | 41.84 | 80.92 | 96.49 |

**Artistic style settings** For artistic style settings, we use the same vocabularies as in the object-related settings, except we exclude ImageNet. For the ChatGPT vocabulary, we modify the prompt to focus on artistic style concepts rather than object concepts. The prompts used are: "Provide k=100

---

[3]https://www.oxfordlearnersdictionaries.com/wordlist/american_english/oxford3000/

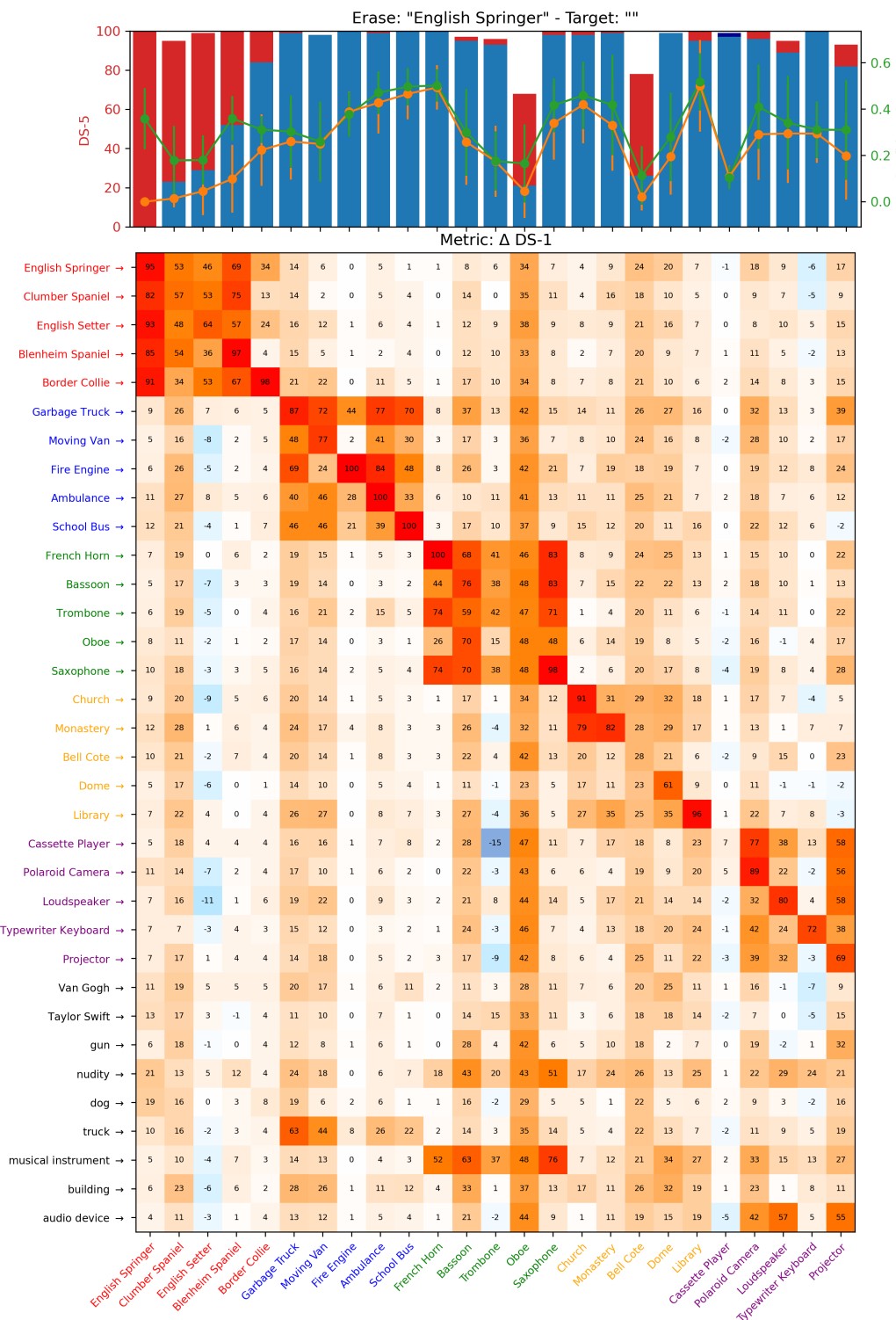

Figure 8: Analysis of the impact of choosing empty concept as the target concept for erasure with DS-1 metric.

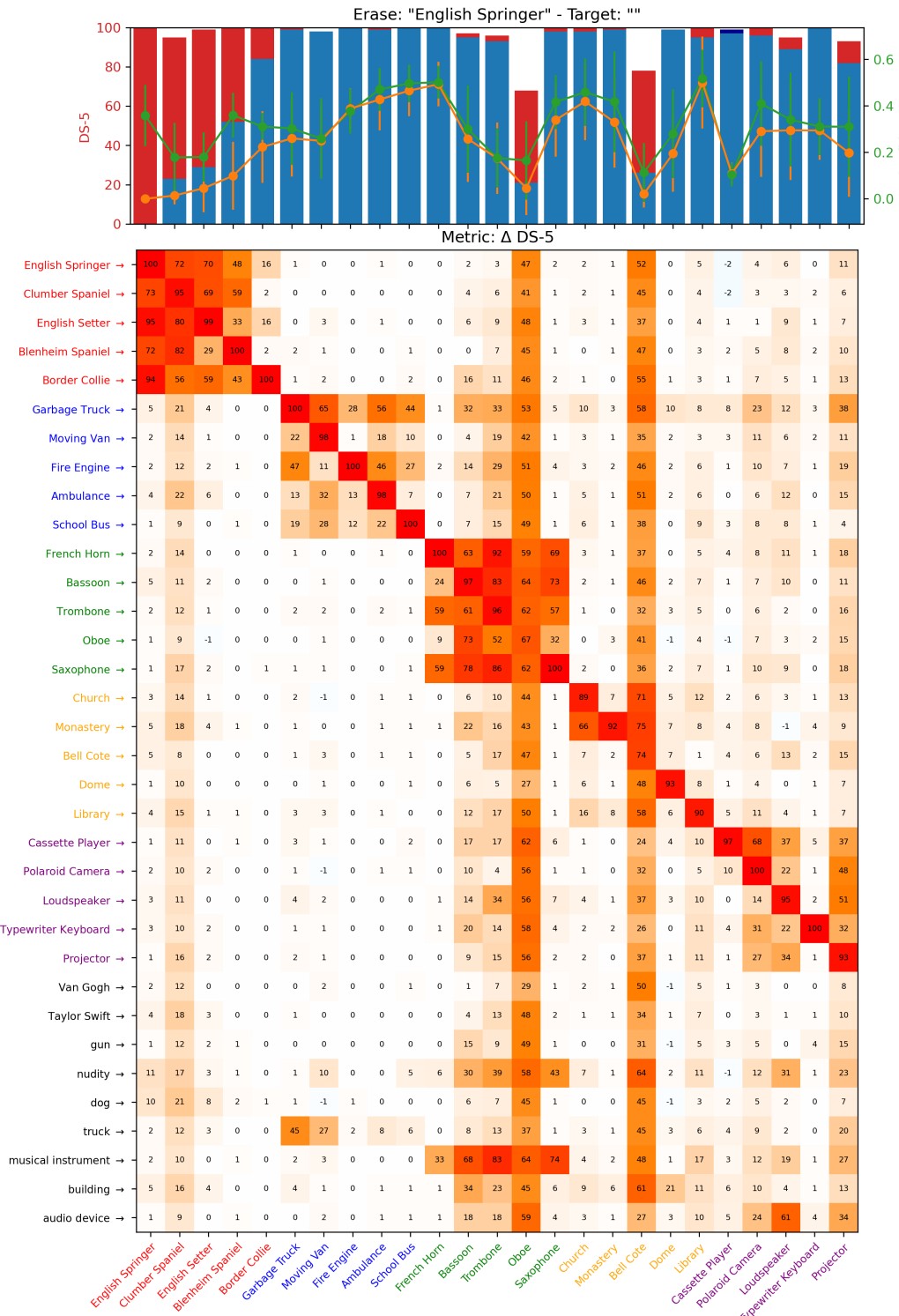

Figure 9: Analysis of the impact of choosing the empty concept as the target concept for erasure with DS-5 metric.

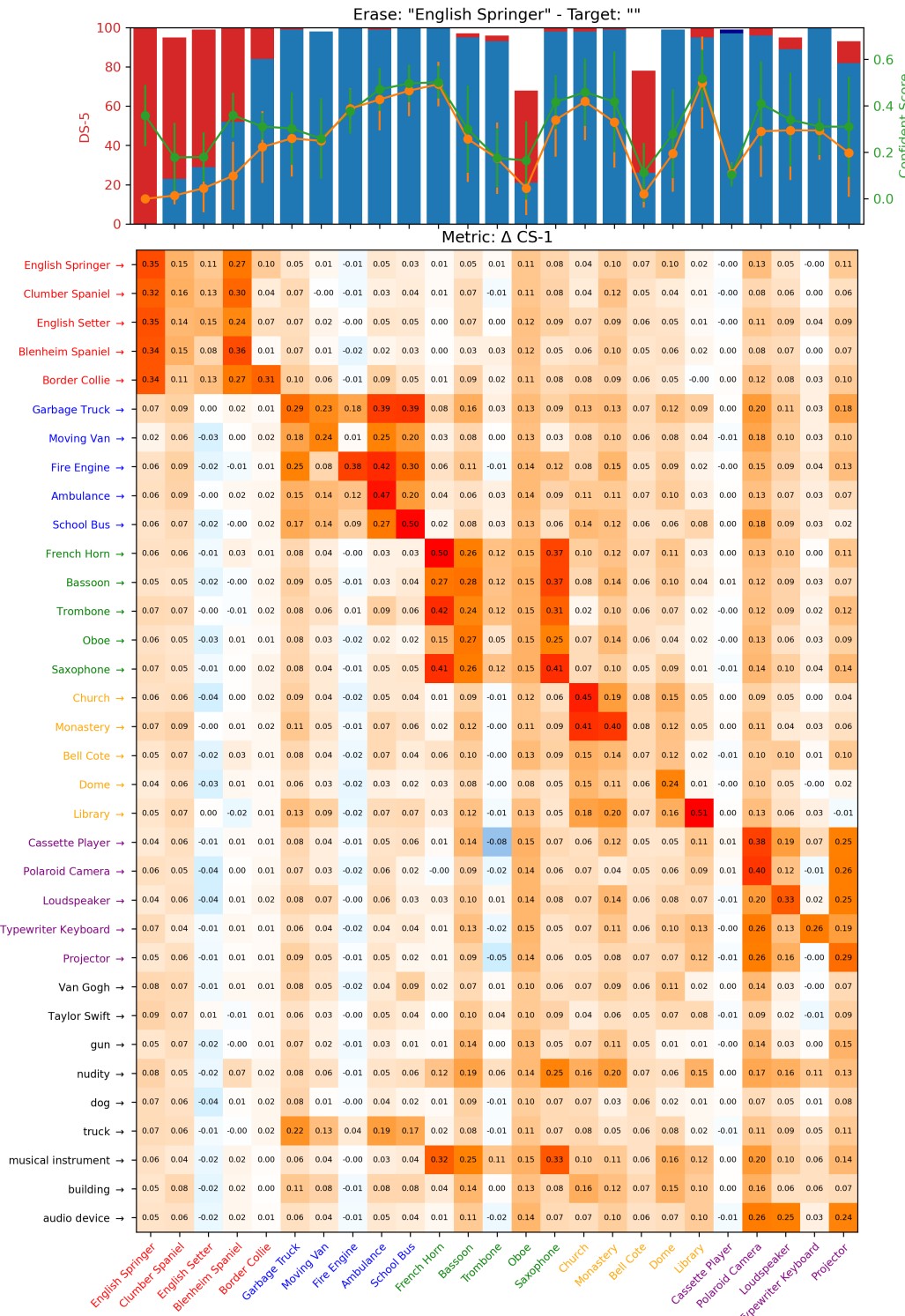

Figure 10: Analysis of the impact of choosing the empty concept as the target concept for erasure with CS-1 metric.

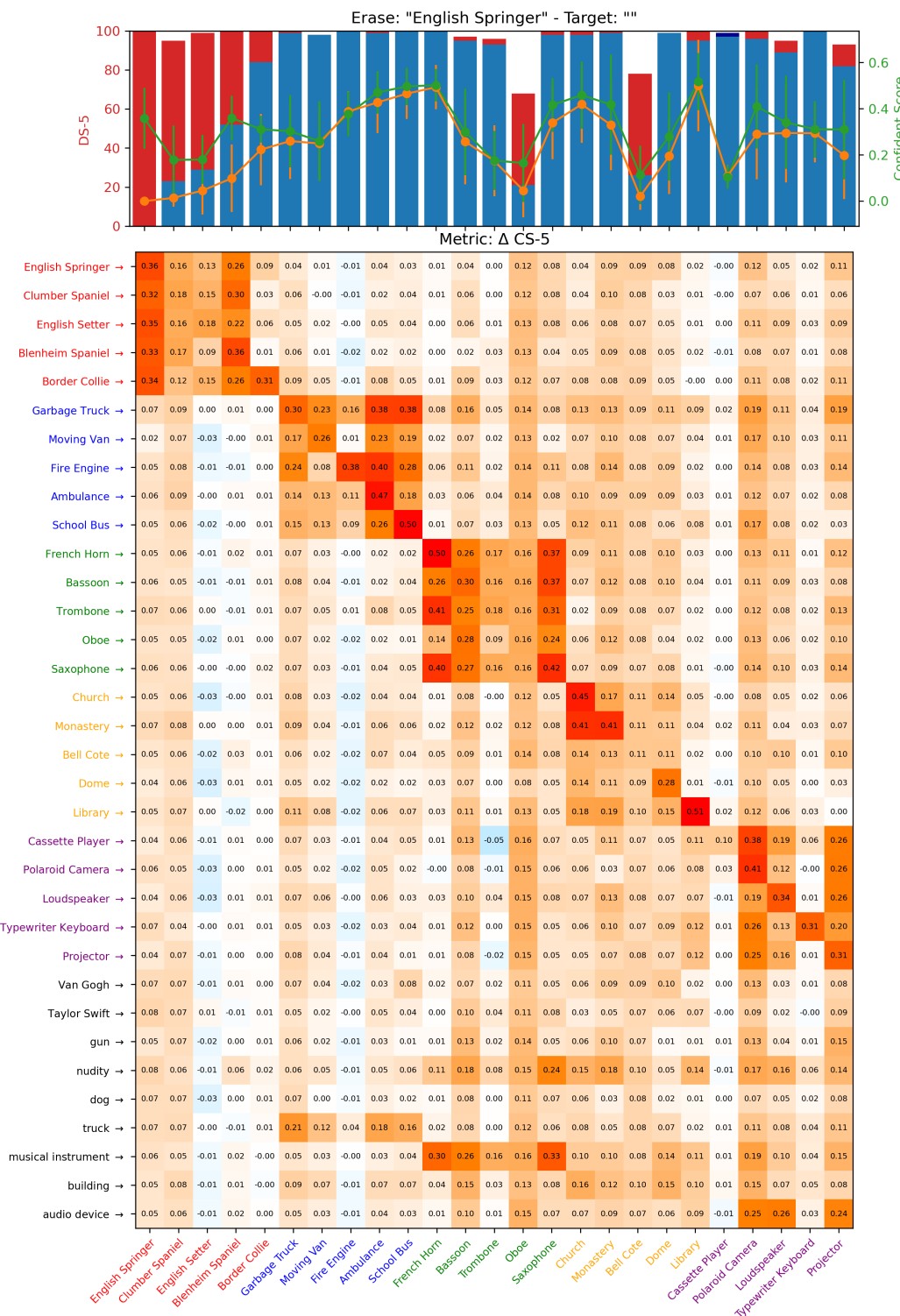

Figure 11: Analysis of the impact of choosing the empty concept as the target concept for erasure with CS-5 metric.

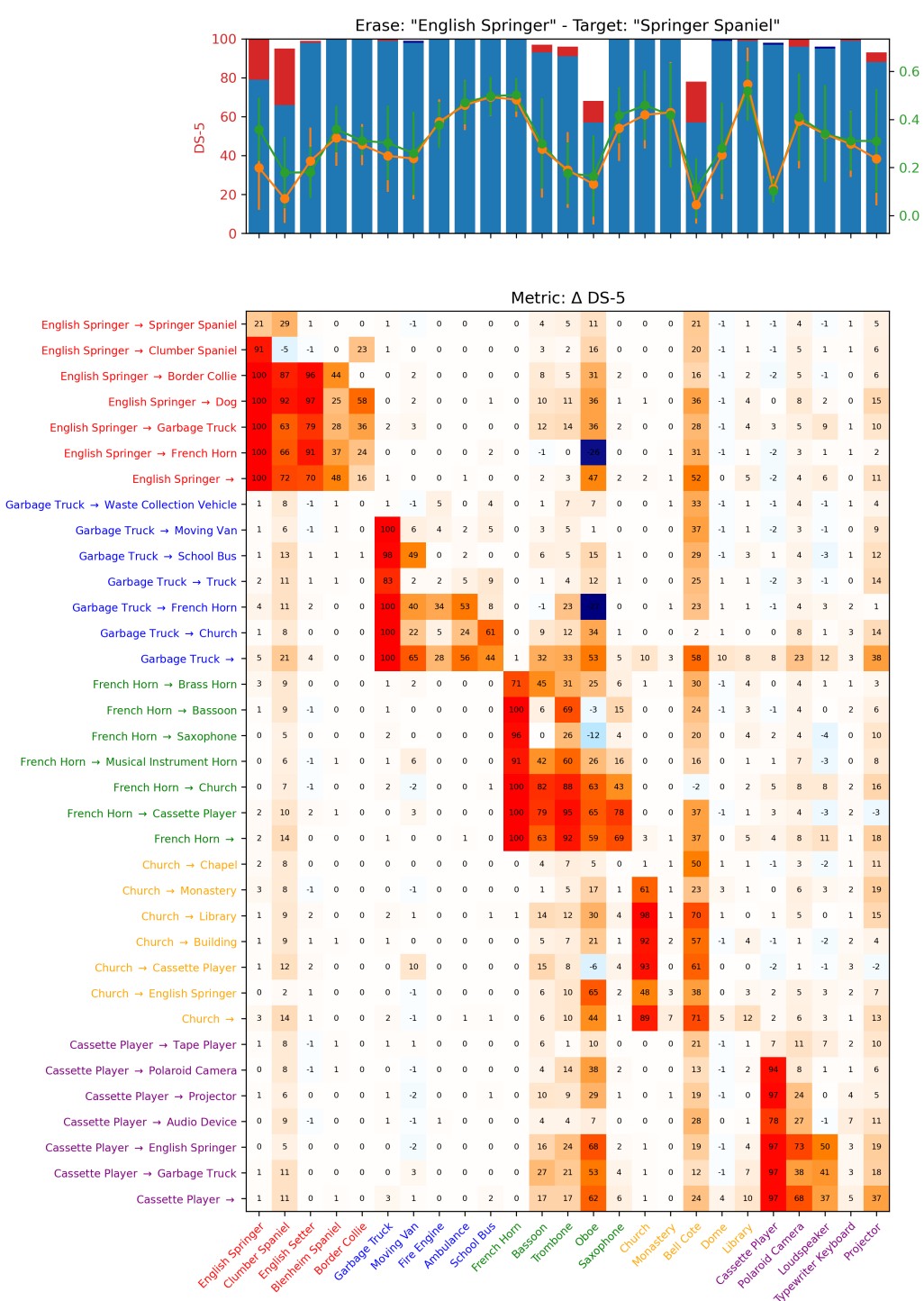

Figure 12: Analysis of the impact of choosing a specific concept as the target concept for erasure with DS-5 metric.

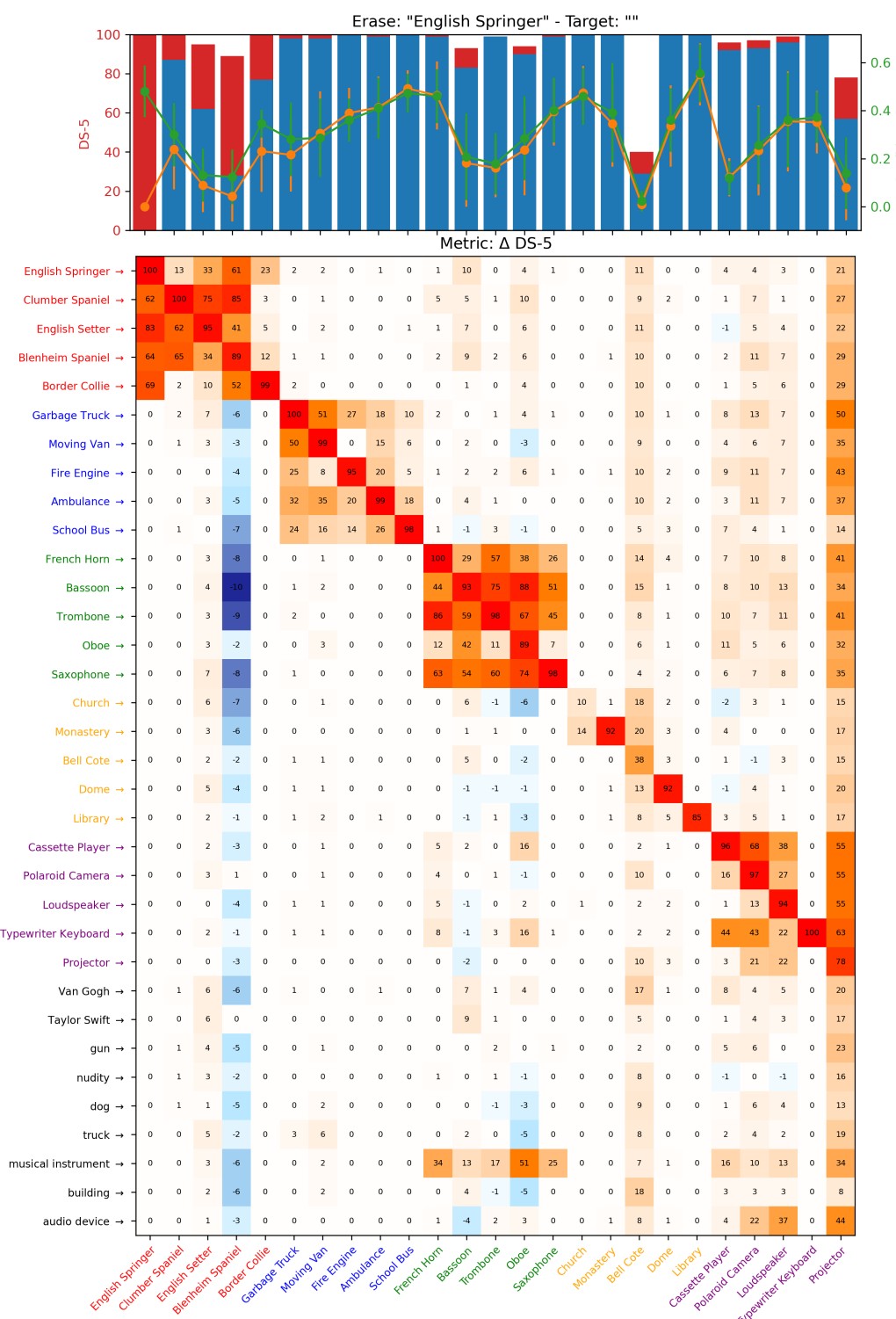

Figure 13: Analysis of the impact of choosing the empty concept as the target concept for erasure with Stable Diffusion version 2 with DS-5 metric.

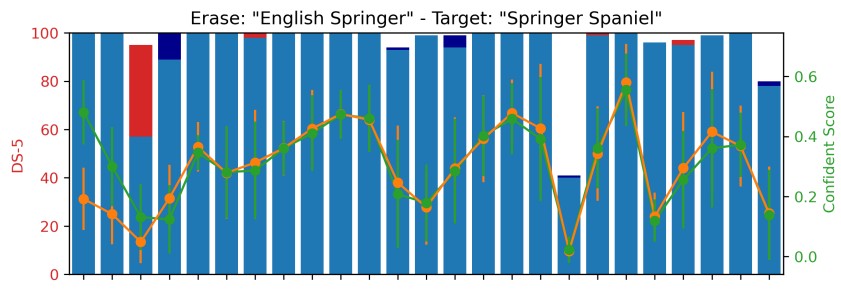

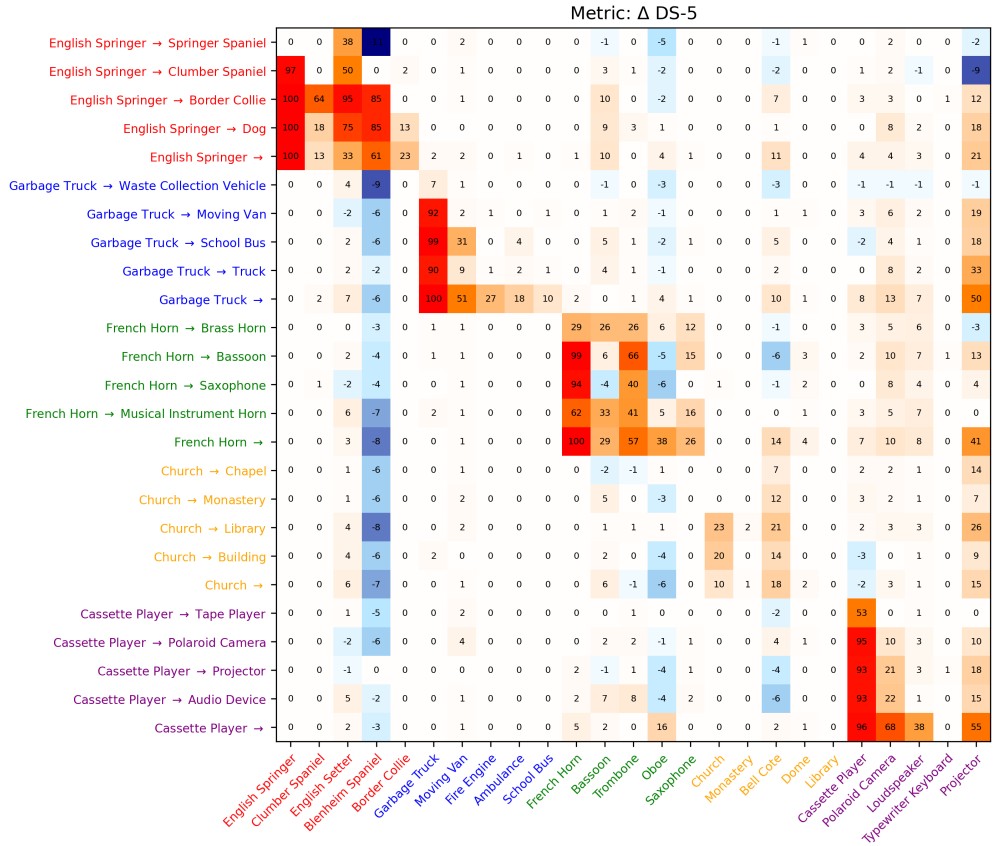

Figure 14: Analysis of the impact of choosing a specific concept as the target concept for erasure with Stable Diffusion version 2 with DS-5 metric.

keywords or phrases that resemble a 'painting,' and export the list to a CSV file" and "Provide k=100 artistic concepts similar to the 'Kelly McKernan' art style, and export the list to a CSV file." The complete list of concepts is available in the corresponding repository. As shown in Table 5, the ChatGPT vocabulary achieves comparable erasing performance to the Oxford vocabulary, while outperforming it in preservation. Based on these results, we select the ChatGPT vocabulary as the default for all artistic style experiments in the main paper.

Table 5: Impact of choosing vocabularies on artistic style settings.

| | To Erase | | To Retain | |
| | CLIP $\downarrow$ | LPIPS $\uparrow$ | CLIP $\uparrow$ | LPIPS $\downarrow$ |
| --- | --- | --- | --- | --- |
| ESD | $23.56 \pm 4.73$ | $0.72 \pm 0.11$ | $29.63 \pm 3.57$ | $0.49 \pm 0.13$ |
| CA | $27.79 \pm 4.67$ | $0.82 \pm 0.07$ | $29.85 \pm 3.78$ | $0.76 \pm 0.07$ |
| UCE | $24.47 \pm 4.73$ | $0.74 \pm 0.10$ | $30.89 \pm 3.56$ | $0.40 \pm 0.13$ |
| Oxford | $22.03 \pm 5.24$ | $0.79 \pm 0.11$ | $29.97 \pm 3.54$ | $0.49 \pm 0.14$ |
| ChatGPT | $22.44 \pm 5.03$ | $0.80 \pm 0.12$ | $30.45 \pm 3.35$ | $0.44 \pm 0.13$ |

Table 6: Erasing artistic style concepts. The best and second-best results are highlighted in bold and underline, respectively.

| | To Erase | | To Retain | |
| | CLIP $\downarrow$ | LPIPS $\uparrow$ | CLIP $\uparrow$ | LPIPS $\downarrow$ |
| --- | --- | --- | --- | --- |
| ESD | $23.56 \pm 4.73$ | $0.72 \pm 0.11$ | $29.63 \pm 3.57$ | $0.49 \pm 0.13$ |
| CA | $27.79 \pm 4.67$ | $\mathbf{0.82 \pm 0.07}$ | $29.85 \pm 3.78$ | $0.76 \pm 0.07$ |
| UCE | $24.47 \pm 4.73$ | $0.74 \pm 0.10$ | $\underline{30.89 \pm 3.56}$ | $\underline{0.40 \pm 0.13}$ |
| MACE | $27.96 \pm 4.22$ | $0.60 \pm 0.10$ | $\mathbf{31.52 \pm 2.91}$ | $\mathbf{0.25 \pm 0.12}$ |
| AP | $\mathbf{21.57 \pm 5.46}$ | $0.78 \pm 0.10$ | $30.13 \pm 3.44$ | $0.47 \pm 0.14$ |
| Ours | $\underline{22.44 \pm 5.03}$ | $\underline{0.80 \pm 0.12}$ | $30.45 \pm 3.35$ | $0.44 \pm 0.13$ |

## D.3 HYPERPARAMETER ANALYSIS

In this experiment, we analyze the impact of the trade-off hyperparameter $\lambda$ and the temperature $\gamma$ in our method. For all other experiments, we fixed $\lambda = 1.0$ and $\gamma = 0.1$ We conduct experiment with object-related setting, erasing five concepts "Cassette Player", "Church", "Garbage Truck", "Parachute", and "French Horn" simultaneously. We vary $\lambda$ and $\tau$ within the range of 0.01 to 2.

It can be seen from Figure 15a that the erasing performance decreases as $\lambda$ increases, while the preserving performance improves monotonically. This aligns with our theoretical analysis in Section 4, where $\lambda$ controls the balance between the erasure loss $L_1$ and the preserving loss $L_2$ in Equation equation 3.

In Figure 15b, we observe that as $\gamma$ increases, the preserving performance declines, though there is no clear trend for erasing performance. It is worth reminding that $\gamma$ is the temperature parameter in the Gumbel-Softmax operator, which controls the discreteness of the mixed weight $G(\pi)$ in Equation equation 4, i.e., the lower $\gamma$ is, the closer $G(\pi)$ is to the true one-hot vector.

## D.4 EVALUATION ON SCALABILITY

We conduct an additional experiment to demonstrate/evaluate the scalability of our proposed method. More specifically, we erase 25 concepts from the NetFive dataset, simultaneously and collect additional 75 other concepts from the ImageNet dataset to form a set of 100 concepts for evaluation. In the preservation set of 75 concepts, we intentionally include 25 concepts that are semantically similar to the 25 concepts being erased and 50 other concepts that are semantically unrelated. We use the ImageNet hierarchy from this website https://observablehq.com/@mbostock/imagenet-hierarchy and Google search to find these visually semantically similar concepts.

Table 7 shows the breakdown of the concepts (to-be-erased and to-be-preserved-similar) used in the experiment.

The results of the experiment are shown in Table 8. We investigate three different vocabularies for the concept space $\mathcal{C}$ in the experiment including the ImageNet (AGE-I), Oxford-3K (AGE-O), and

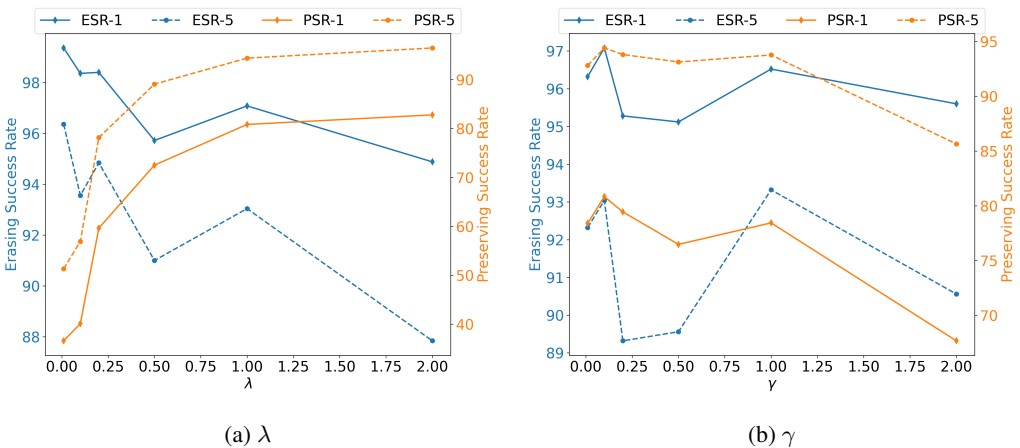

(a) $\lambda$          (b) $\gamma$

Figure 15: Ablation study on the trade-off hyperparameter $\lambda$ and the temperature hyperparameter $\gamma$.

| Super-Category | To-be-erased | To-be-preserved |
|---|---|---|
| Dog | English Springer, Clumber Spaniel, English Setter, Blenheim Spaniel, Border Collie | Chihuahua, Tibetan Mastiff, Red Fox, White Wolf, Hyena |
| Vehicle | Garbage Truck, Moving Van, Fire Engine, Ambulance, School Bus | Moped, Model T, Golf Cart, Tractor, Forklift |
| Music Instrument | French Horn, Bassoon, Trombone, Oboe, Saxophone | Organ, Grand Piano, Guitar, Drum, Cello |
| Building | Church, Monastery, Bell Cote, Dome, Library | Boathouse, Greenhouse, Cinema, Bookshop, Restaurant |
| Device | Cassette Player, Polaroid Camera, Loudspeaker, Typewriter Keyboard, Projector | Cellular Telephone, Laptop, Television, Desktop Computer, iPod |

Table 7: Breakdown of concepts used in scalability experiment

the manually crafted vocabulary (AGE-M), where we leverage the knowledge of the to-be-erased concepts to generate the vocabulary, i.e., which words are semantically related to the to-be-erased concepts, like "dog", "car", "instrument", etc. This manually crafted vocabulary is similar to the Oxford-3K but much smaller in size. We compare the performance of AGE with the ESD and UCE methods.

Compared to the baseline methods, it can be seen that UCE totally fails when the number of concepts increases. Our AGE method with Oxford-3K vocabulary (AGE-O) outperforms the ESD method in both erasure and preservation performance. Compared among the three vocabularies, it can be seen that while AGE-O outperforms other vocabularies in erasure performance, it has significant performance drop in preservation performance. The AGE method with manually crafted vocabulary (AGE-M) achieves the best trade-off between erasure and preservation performance, which has a small drop in erasure performance compared to ESD but a much better preservation performance, which is consistent with our analysis in Section 3.2.

### D.5 MIXTURE OF CONCEPTS

**Why mixture of concepts?** Given that the concept space $\mathcal{C}$ is discrete and finite, a natural approach to solving Equation equation 3 would be to enumerate all the concepts in $\mathcal{C}$ and select the most sensitive concept $c_t$ that maximizes the total loss at each optimization step of the outer minimization

| Concept | SD | ESD | UCE | AGE-I | AGE-O | AGE-M |
|---|---|---|---|---|---|---|
| *Erased Concepts* | | | | | | |
| dog | 99.4 | 11.6 | 0.0 | 35.8 | 11.8 | 7.6 |
| truck | 99.4 | 23.4 | 0.0 | 14.4 | 9.8 | 6.4 |
| inst. | 98.6 | 10.8 | 0.0 | 28.8 | 12.8 | 16.0 |
| build. | 91.2 | 38.8 | 0.0 | 61.2 | 53.0 | 74.8 |
| elect. | 95.0 | 23.8 | 0.0 | 31.6 | 11.8 | 52.0 |
| *Similar Concepts* | | | | | | |
| dog | 100.0 | 94.4 | 0.0 | 99.8 | 92.2 | 99.2 |
| truck | 100.0 | 79.4 | 0.0 | 99.4 | 78.6 | 96.4 |
| inst. | 98.8 | 33.4 | 0.0 | 80.4 | 34.8 | 84.8 |
| build. | 97.4 | 77.0 | 0.0 | 90.6 | 86.6 | 95.2 |
| elect. | 92.0 | 21.6 | 0.0 | 69.6 | 35.6 | 80.0 |
| *General Concepts* | | | | | | |
| mamm. | 99.8 | 98.2 | 0.0 | 99.8 | 98.4 | 100.0 |
| bird | 99.6 | 87.0 | 0.2 | 98.4 | 86.6 | 97.2 |
| rept. | 95.6 | 83.2 | 0.0 | 94.4 | 83.8 | 89.2 |
| insect | 78.6 | 66.4 | 0.0 | 75.8 | 65.0 | 74.4 |
| fish | 93.8 | 73.0 | 0.0 | 95.8 | 64.6 | 92.8 |
| veh. | 99.8 | 82.8 | 0.0 | 98.8 | 80.0 | 98.0 |
| craft | 99.2 | 64.2 | 0.0 | 96.6 | 70.4 | 94.4 |
| furn. | 96.8 | 64.6 | 0.4 | 83.0 | 72.0 | 80.4 |
| fruit | 100.0 | 81.6 | 0.0 | 99.8 | 83.8 | 99.2 |
| obj. | 100.0 | 74.4 | 0.2 | 94.8 | 76.8 | 95.6 |
| ESR-1 | 24.2 | 90.6 | 100.0 | 84.7 | 91.2 | 86.1 |
| ESR-5 | 3.3 | 78.3 | 100.0 | 65.6 | 80.2 | 68.6 |
| PSR-1 | 83.1 | 56.3 | 0.0 | 74.7 | 57.1 | 73.5 |
| PSR-5 | 96.8 | 72.1 | 0.1 | 91.8 | 73.9 | 91.8 |

Table 8: Comparison of different methods on concept erasure.

problem. However, this method is computationally impractical due to the large number of concepts in $\mathcal{C}$. Moreover, many concepts are inherently complex, often composed of multiple attributes, and cannot always be interpreted as singular, isolated concepts within the space $\mathcal{C}$.

Since the concepts are represented textually and the output of Latent Diffusion Models is controlled by textual prompts, the most intuitive way to create a mixture of concepts, such as combining $c_1$ and $c_2$, is through a textual template. For example, using a prompt like "a photo of a $c_1$ and a $c_2$" ensures that both concepts appear in the generated image. However, this approach has a significant limitation: it does not provide gradients, preventing us from using standard backpropagation to learn and fine-tune the target concept $c_t$.

To address this issue, we employ the Gumbel-Softmax operator to approximate a mixture of concepts. We set the temperature to a value below 1, ensuring that the resulting target concept is a combination of a few dominant concepts, rather than an indiscriminate mixture of all the concepts in $\mathcal{C}$.

**Visualization** To illustrate the effect of concept mixtures as described in Equation equation 4, we visualize the generated images $g(z_T, (1 - \alpha)\tau(c_1) + \alpha\tau(c_2))$ in Figure 16, where $g()$ represents the image generation function (the diffusion backward process), $z_T$ is the initial noise input, and $\tau$ is the textual encoder. We fix $c_1$ as "English Springer" and vary $\alpha$ from 0 to 1 with different $c_2$ values to simulate the mixture of concepts $G(\pi) \cdot \mathcal{C}$ in Equation equation 4. Intuitively, we expect the generated images to gradually transition from concept $c_1$ to concept $c_2$ as $\alpha$ increases.

In cases like "Church," "French Horn," and "Garbage Truck," this gradual transformation is indeed observable, where the image transitions smoothly from "English Springer" to the target concept as $\alpha$ increases. However, for other concepts like "Border Collie" and "Golf Ball," the transformation is less smooth. For instance, the image shifts abruptly from "English Springer" to "Border Collie"

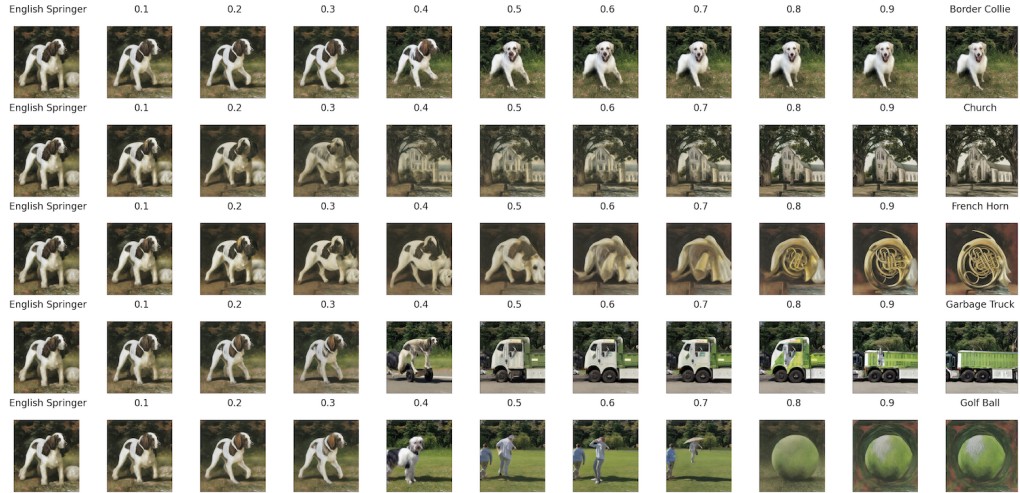

Figure 16: Visualization of the output images $g(z_0, (1-\alpha)\tau(c_1) + \alpha\tau(c_2))$ with different $\alpha$ and $c_2$.

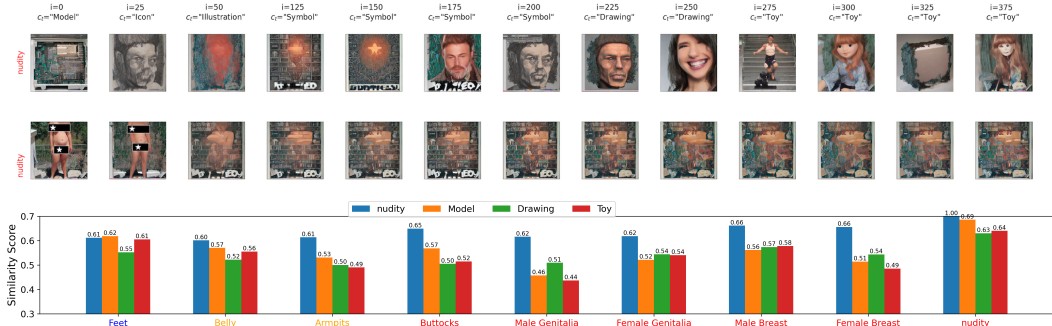

Figure 17: Top/a: Intermediate results of the search process, with images generated from the most sensitive concepts $c_t$ found by our method and $c_e$ at the same optimization step. Bottom/b: Similarity between nudity attributes and keywords. The figure is repeated for reading convenience.

when $\alpha$ increases from 0.4 to 0.5, or from "English Springer" to "Golf Ball" as $\alpha$ changes from 0.4 to 0.7.

This phenomenon suggests that mixing concepts is not merely a linear interpolation between two target concepts, but is influenced by the intrinsic nature of the concepts $c_1$ and $c_2$.

## D.6 Searching for the optimal target concept

**Erasing Nudity Concept** We investigate the search for optimal target concepts $c_t$ by visualizing the generated images from $c_t$ found by our method and the corresponding output of $c_e$ at the same optimization step as shown in Figure 17a. Additionally, we present the similarity scores between the nudity attributes detected by the detector and $c_t$ as in Figure 17b. Firstly, the to-be-erased concept $c_e$ ("nudity") is closely related to the sensitive body parts such as "breasts" and "genitalia", explaining why removing the "nudity" concept also eliminates these sensitive attributes. On the other hand, as discussed in Section 4, the target concept $c_t$ is designed to be locally aligned with $c_e$ but not exactly the same as $c_e$. In our results, all intermediate concepts $c_t$ such as "Model", "Drawing", and "Toy" are highly correlated with the "nudity" concept satisfying the first condition. However, while being highly correlated with less sensitive parts such as "Feet", they are less correlated with more sensitive ones like "breasts" and "genitalia", meeting the second condition. It is a worth recall that in Equation 3, $c_t$ serves as a retained concept to preserve the model's capabilities. Therefore, the strong correlation between $c_t$ and "Feet", and its weaker connection with other sensitive parts, explains the interesting advantage of our method that it can still retain the "Feet" concept while successfully erasing others, as observed in Figure 3.

**Erasing Object-Related Concepts**   To further illustrate how our method operates across different settings, we provide intermediate results of the search process in Figures 18, 19. These experiments follow the same setup as in Section 5.1, exploring the effect of using various vocabularies including CLIP, Oxford, ChatGPT, and ImageNet, as introduced in Section D.2.

In each subfigure, the first row depicts images generated from the most sensitive concepts $c_t$ identified by our method, while the second row shows the corresponding to-be-erased concepts $c_e$. Importantly, all images are generated from the same initial noise input $z_T$, resulting in similar backgrounds, while still featuring the relevant concepts $c_t$ and $c_e$.

The results reveal that different vocabularies lead to distinct erasing outcomes.

It can be seen that different vocabularies lead to different erasing effects. For instance, when using ChatGPT's vocabulary, as shown in Figure 18c, the intermediate concepts $c_t$ identified include ["Chapel," "Altar room," "Shrine"], which are semantically similar to the to-be-erased concept "Church." This close semantic similarity results in a weaker erasing effect, where the "Church" concept is not fully removed in the generated images. This phenomenon is also evident in other object-related concepts, such as "Garbage Truck" as shown in Figure 19c. This result is consistent with the low erasing performance score observed in Table 4 as discussed in Section D.2.

In contrast, using ImageNet as the vocabulary, as shown in Figures 18d and 19d, leads to more effective erasure. Initially, the target concepts $c_t$ are closely related to the to-be-erased concepts $c_e$, but they gradually shift to less related concepts. This results in a stronger erasing effect, as outlined in Section D.2.

### D.7    IMPACT OF ERASING ON SYNONYMS

Given the list of 'valid' synonyms as introduced in Section C.3, we generated images using the sanitized models from four object-related settings, where each setting corresponds to erasing five Imagenette concepts simultaneously, as described in Section 5.1. We then evaluated the sanitized models using the ESR-1, ESR-5, PSR-1, and PSR-5 metrics. Specifically, if an image generated from a synonym of a to-be-erased concept is classified as that concept, the model has failed to erase it, and vice versa. The results are shown in Table 9.

First, the SD-org and SD-syn represent the results of the original model using the original concepts and their synonyms, respectively. Notably, the PSR-1 score for SD-syn is only 58.5%, indicating that out of 100 images generated from synonyms of a concept like "Trash Truck," only 58.5% are classified as "Garbage Truck" by the ResNet-50 model. This is significantly lower than the 78% PSR-1 of the original concept (SD-org). However, for PSR-5, SD-syn achieves 92.5%, which is only 5% lower than SD-org. This suggests that while the images generated from synonyms may not be recognized as the target concept in the top-1 prediction, they are still often classified correctly within the top-5.

With this understanding, we analyzed the performance of erasure methods in handling synonyms of both the to-be-erased and to-be-preserved concepts.

The results show that while UCE and ESD maintain strong erasure performance for the to-be-erased concepts, they struggle to preserve the synonyms of the to-be-preserved concepts. Conversely, CA achieves better preservation but at the cost of reduced erasure performance. MACE strikes a better balance between erasing and preserving, making it the most effective baseline. Our method outperforms MACE in preserving performance by a large margin, achieving the best results overall, though MACE still leads in erasure effectiveness.

### D.8    QUALITATIVE RESULTS

This section provides qualitative examples to further highlight the effectiveness of our approach in comparison to the baselines. Due to internal policies regarding sensitive content, we are only able to display results from two settings: erasing object-related concepts and erasing artistic concepts.

**Erasing Object-Related Concepts**   Figures 21, 22, and 23 show the results of erasing object-related concepts using ESD, UCE, and our method, respectively. Figure 20 shows the generated

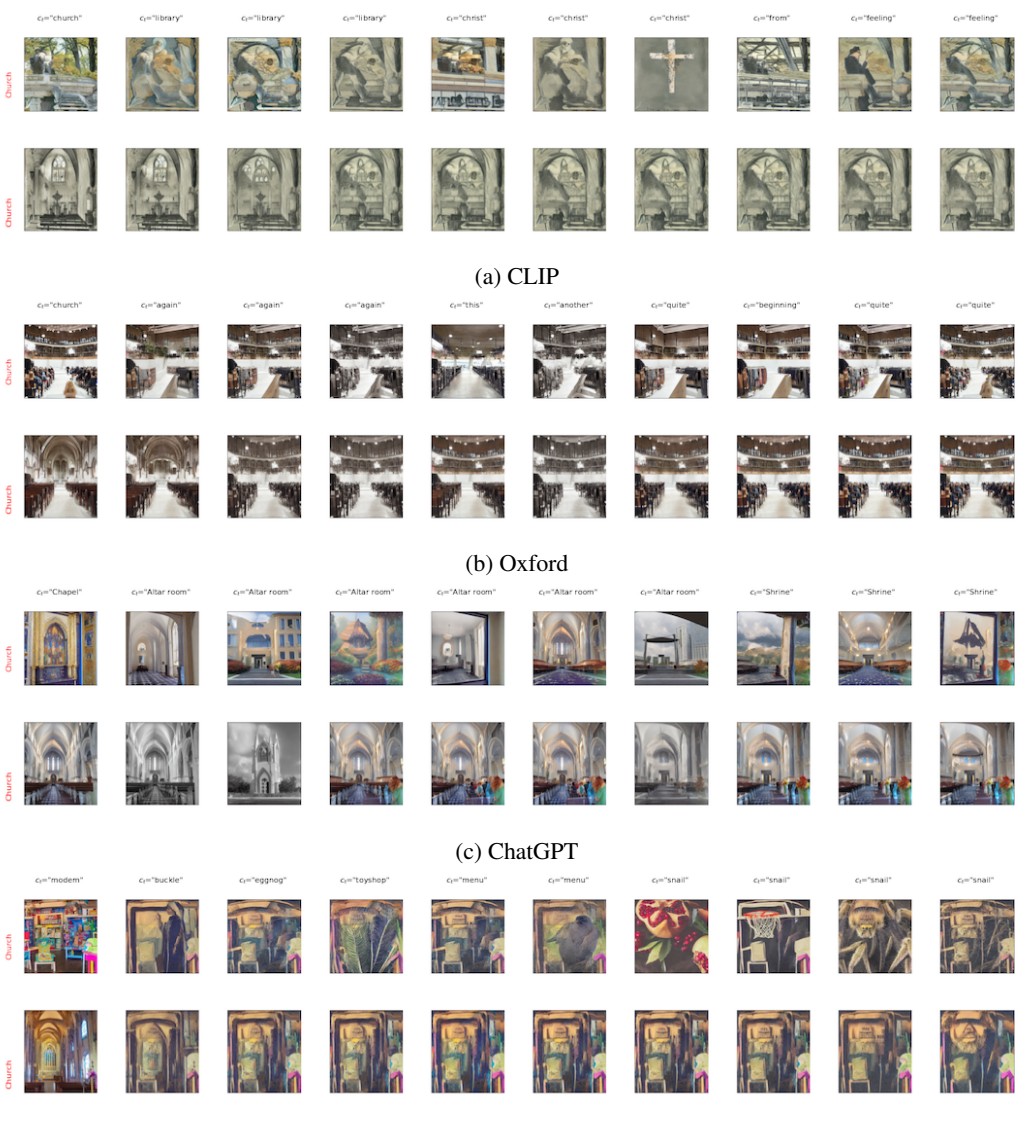

Figure 18: Intermediate results of the search process. The first row is generated from the most sensitive concepts $c_t$ found by our method, and the second row is generated from the corresponding to-be-erased concepts $c_e$ "Church". Each column represents different fine-tuning steps in increasing order. Each subfigure represents for different vocabularies.

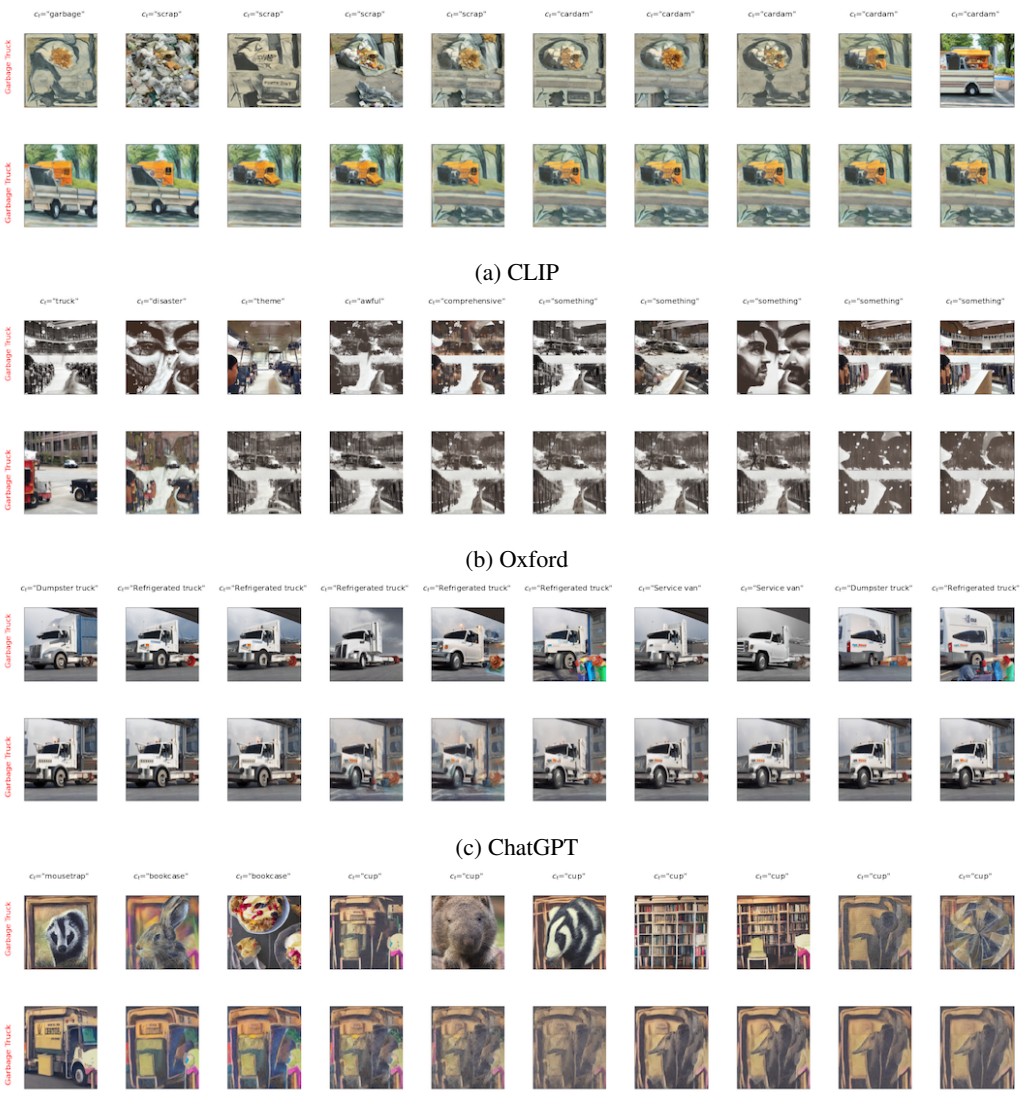

Figure 19: Intermediate results of the search process. The first row is generated from the most sensitive concepts $c_t$ found by our method, and the second row is generated from the corresponding to-be-erased concepts $c_e$ "Garbage Truck". Each column represents different fine-tuning steps in increasing order. Each subfigure represents for different vocabularies.

Table 9: Impact of erasing on synonyms

| Method | ESR$_s$-1↑ | ESR$_s$-5↑ | PSR$_s$-1↑ | PSR$_s$-5↑ |
|--------|-----------|-----------|-----------|-----------|
| SD-org | $22.0 \pm 11.6$ | $2.4 \pm 1.4$ | $78.0 \pm 11.6$ | $97.6 \pm 1.4$ |
| SD-syn | $41.5 \pm 8.2$ | $7.5 \pm 2.3$ | $58.5 \pm 8.2$ | $92.5 \pm 2.3$ |
| ESD | $91.5 \pm 0.5$ | $81.6 \pm 2.5$ | $33.4 \pm 7.3$ | $62.8 \pm 6.6$ |
| CA | $84.1 \pm 2.2$ | $67.4 \pm 5.3$ | $46.2 \pm 7.4$ | $80.3 \pm 1.9$ |
| UCE | $99.8 \pm 0.1$ | $99.2 \pm 0.5$ | $19.5 \pm 4.4$ | $43.8 \pm 0.6$ |
| MACE | $98.1 \pm 0.9$ | $84.7 \pm 2.2$ | $41.4 \pm 10.3$ | $73.3 \pm 3.1$ |
| Ours | $78.8 \pm 6.3$ | $62.3 \pm 7.2$ | $55.4 \pm 8.3$ | $90.0 \pm 2.5$ |

images from the original SD model. Each column represents different random seeds, and each row displays the generated images from either the to-be-erased objects or the to-be-preserved objects.

From Figure 20, we can see that the original SD model can generate all objects effectively. When erasing objects using ESD (Figure 21), the model maintains the quality of the preserved objects, but it also generates objects that should have been erased, such as the "Church" in the second row. This aligns with the quantitative results in Table 1, where ESD achieves the lowest erasing performance.

When using UCE (Figure 22), the model effectively erases the objects as shown in rows 1-5, but the quality of the preserved objects is significantly degraded, such as "tench" and "English springer" in the 8th and 9th rows. This is consistent with the quantitative results in Table 1, where UCE achieves the highest erasing performance but the lowest preservation performance.

In contrast, our method (Figure 23) effectively erases the objects while maintaining the quality of the preserved objects.

**Erasing Artistic Concepts**   Figures 24, 25 show the results of erasing artistic style concepts using our method compared to the baselines. Each column represents the erasure of a specific artist, except the first column, which represents the generated images from the original SD model. Each row displays the generated images from the same prompt but with different artists. The ideal erasure should result in changes in the diagonal pictures (marked by a red box) compared to the first column, while the off-diagonal pictures should remain the same. The results demonstrate that our method effectively erases the artistic style concepts while maintaining the quality of the remaining concepts.

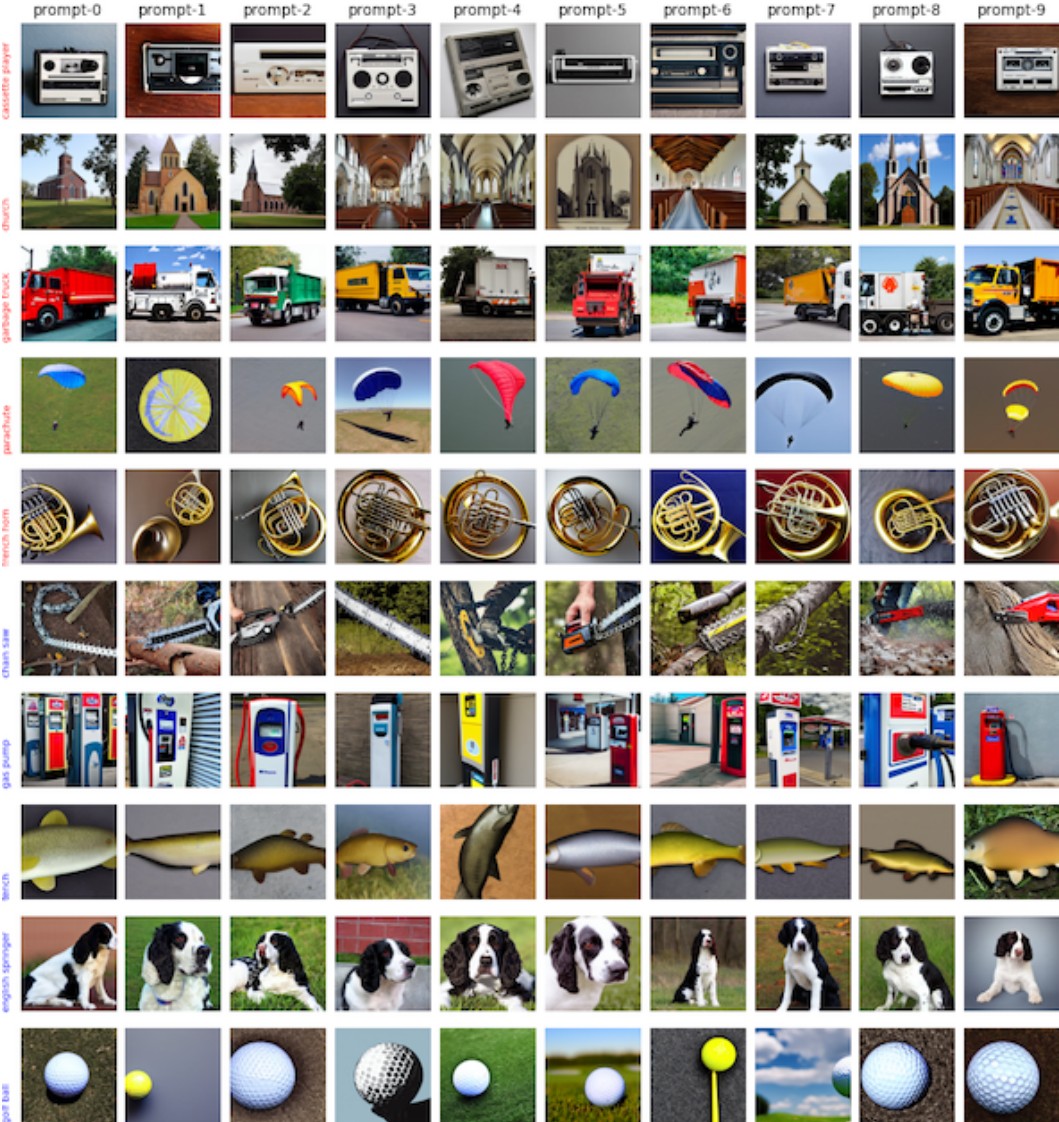

Figure 20: Generated images from the original model. Five first rows are to-be-erased objects (marked by red text) and the rest are to-be-preserved objects. Each column represents different random seeds.

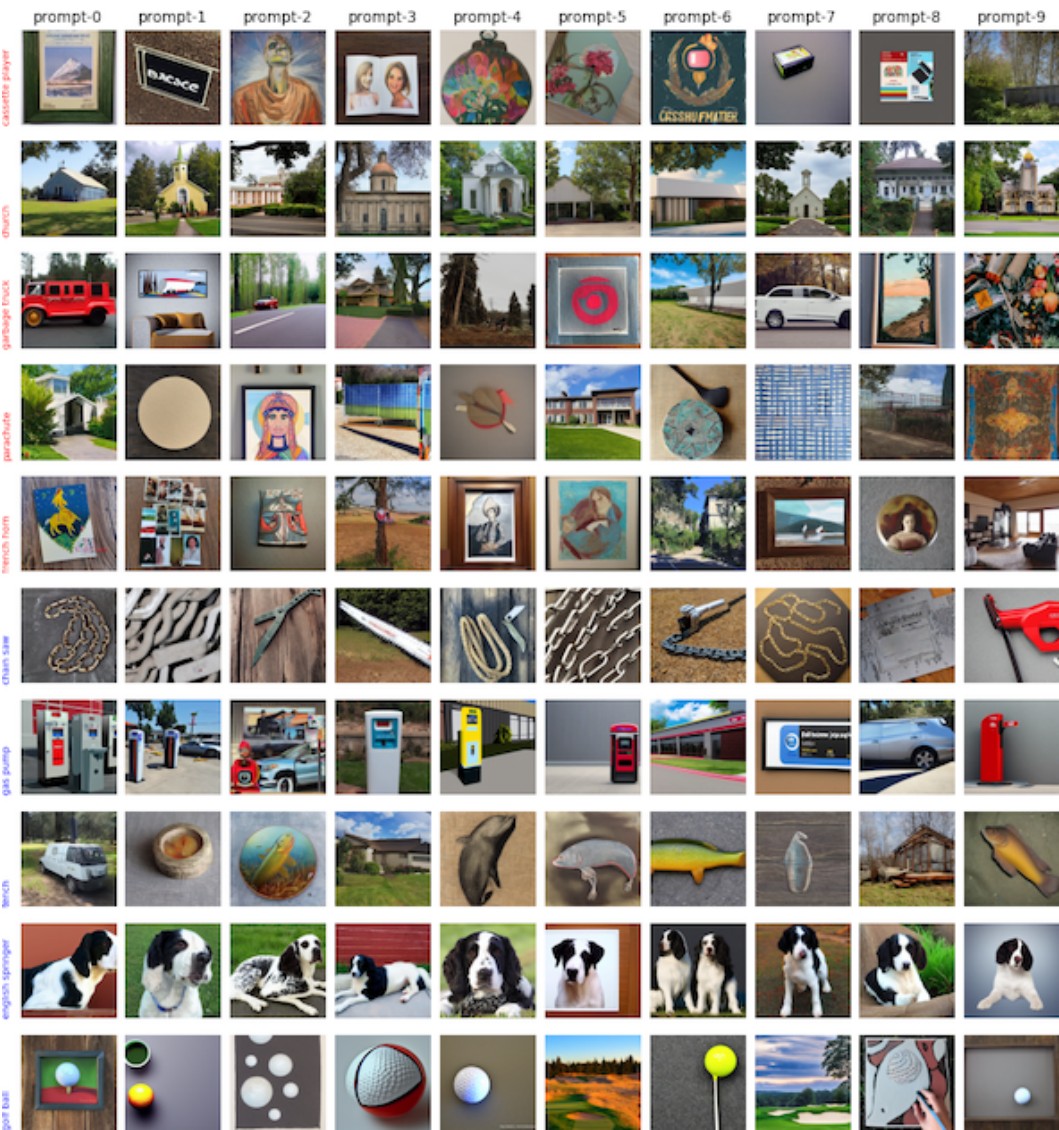

Figure 21: Erasing objects using ESD. Five first rows are to-be-erased objects (marked by red text) and the rest are to-be-preserved objects. Each column represents different random seeds.

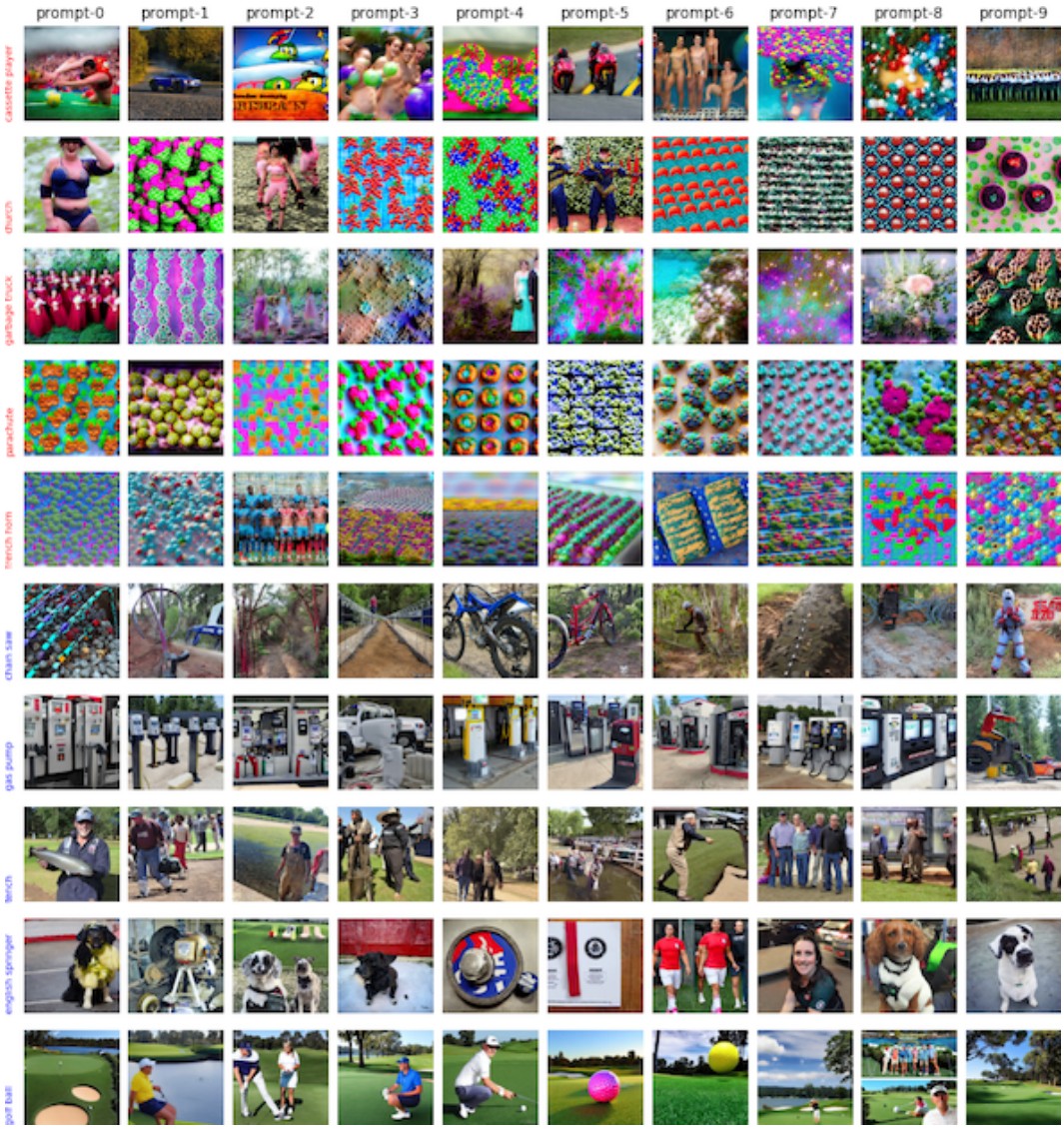

Figure 22: Erasing objects using UCE. Five first rows are to-be-erased objects (marked by red text) and the rest are to-be-preserved objects. Each column represents different random seeds.

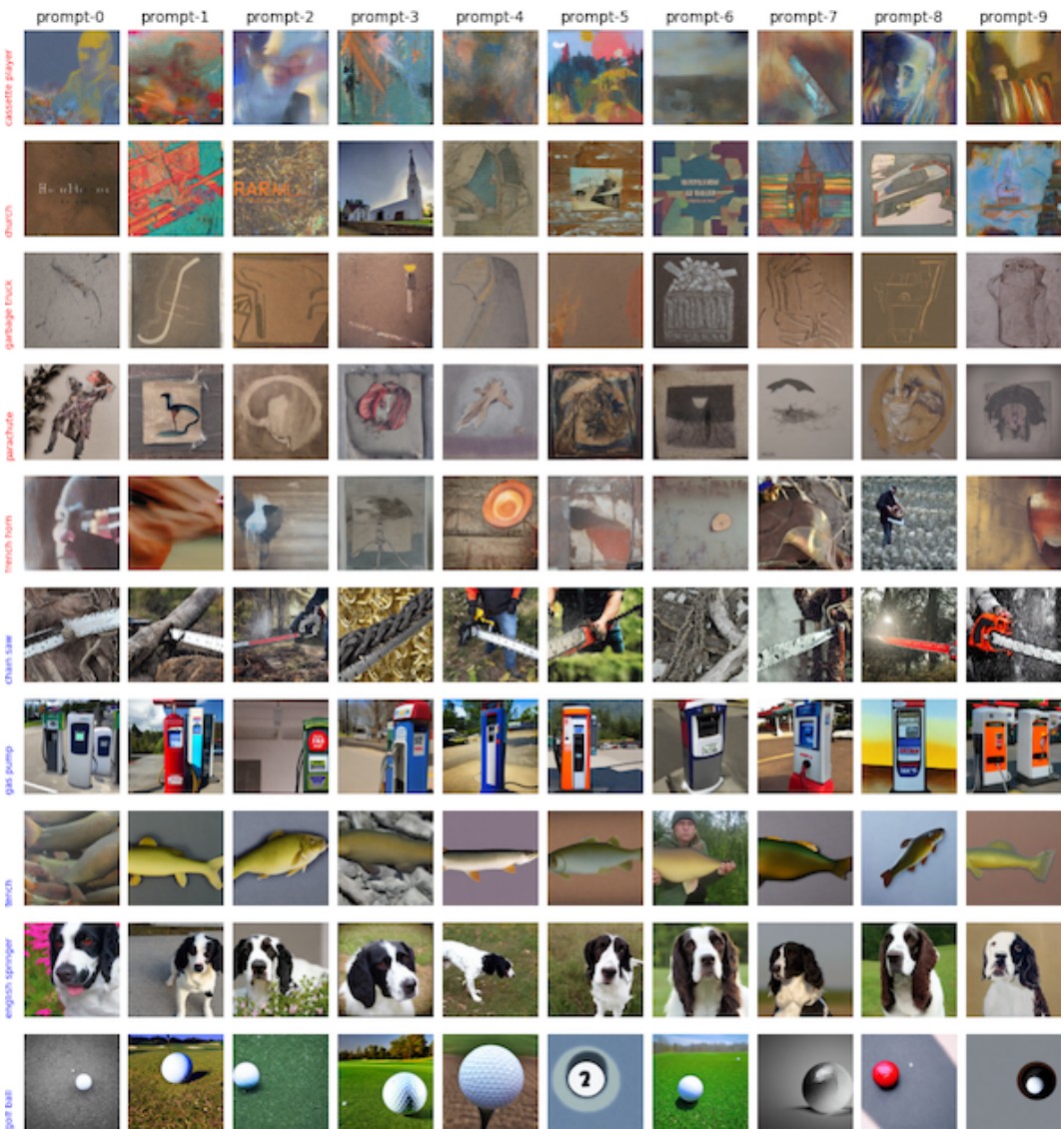

Figure 23: Erasing objects using our method. Five first rows are to-be-erased objects (marked by red text) and the rest are to-be-preserved objects. Each column represents different random seeds.

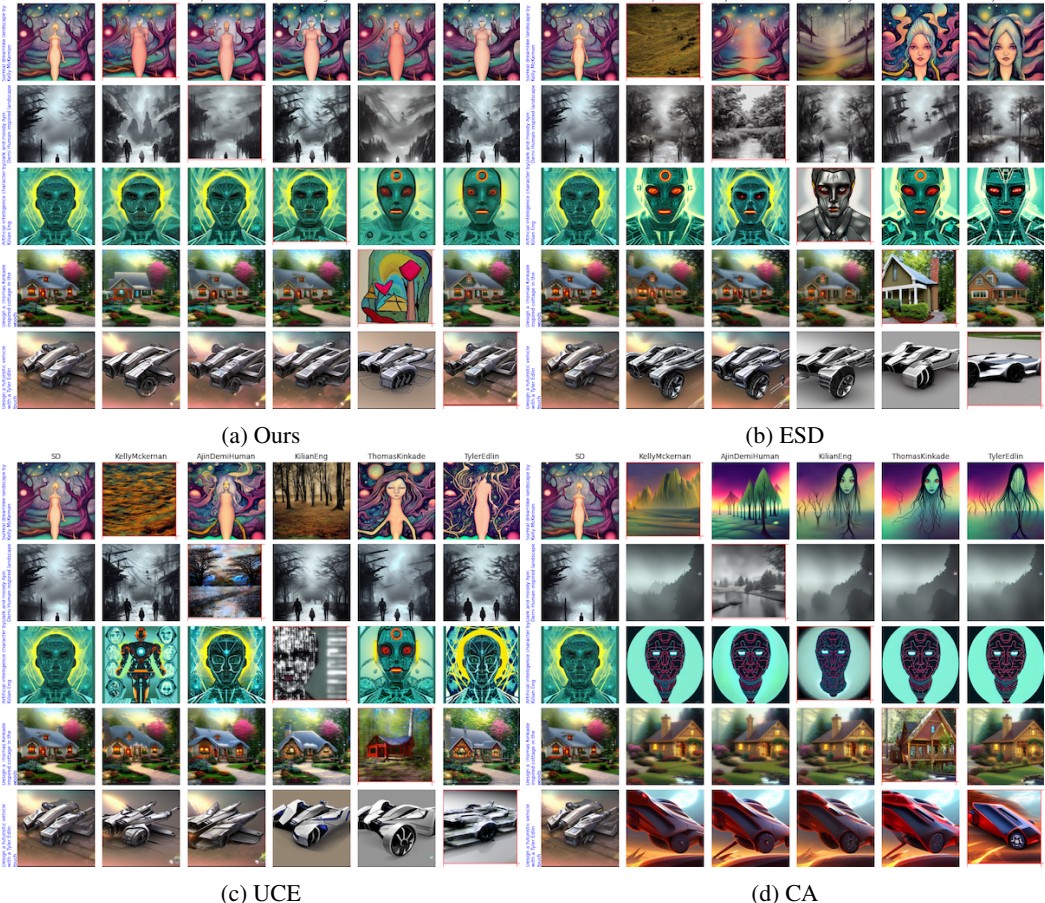

Figure 24: Erasing artistic style concepts. Each column represents the erasure of a specific artist, except the first column which represents the generated images from the original SD model. Each row represents the generated images from the same prompt but with different artists. The ideal erasure should result in the change in the diagonal pictures (marked by a red box) compared to the first column, while the off-diagonal pictures should remain the same.

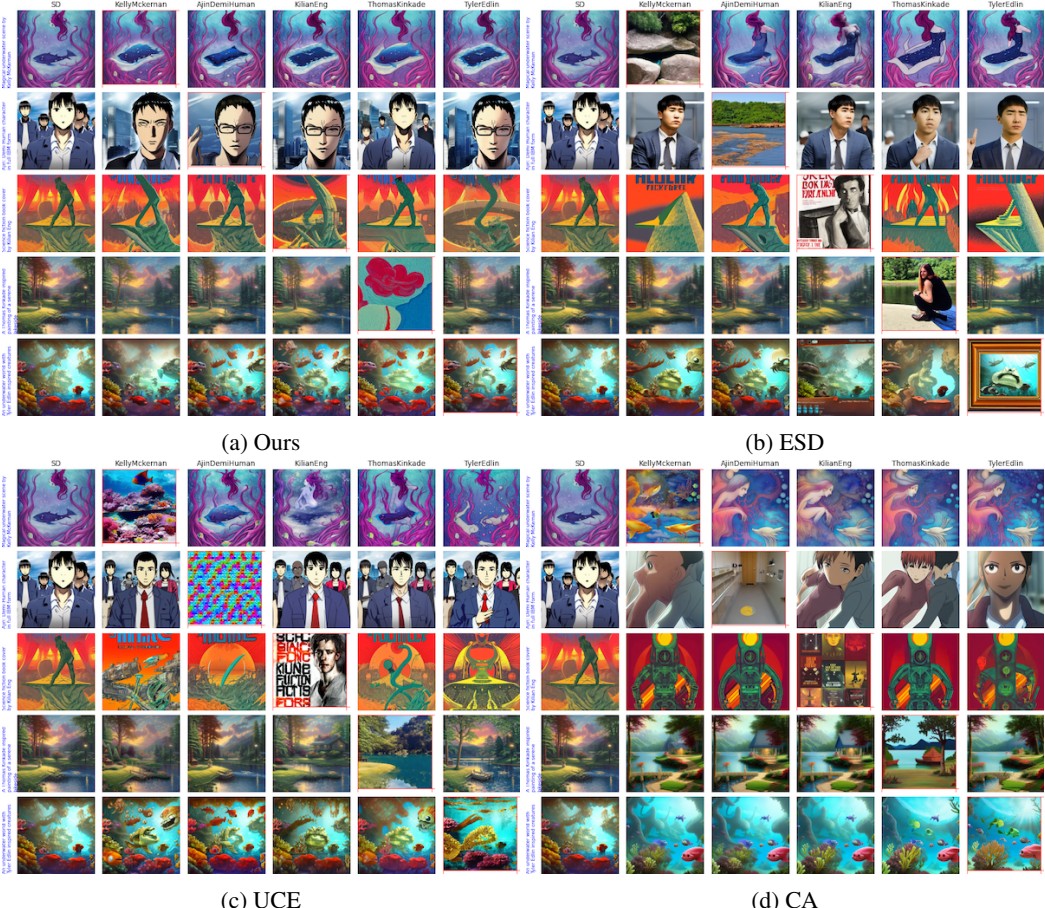

Figure 25: Erasing artistic style concepts (continued). Each column represents the erasure of a specific artist, except the first column which represents the generated images from the original SD model. Each row represents the generated images from the same prompt but with different artists. The ideal erasure should result in the change in the diagonal pictures (marked by a red box) compared to the first column, while the off-diagonal pictures should remain the same.

