# OpenReview forum: "Fantastic Targets for Concept Erasure in Diffusion Models and Where To Find Them"
_ICLR.cc/2025/Conference — ICLR 2025 Poster_

### Official Review · Reviewer_BfS9 · 2024-11-01

**Soundness:** 2
**Presentation:** 3
**Contribution:** 2
**Rating:** 5
**Confidence:** 2

**Summary:**

This paper introduces the AGE method, aimed at enhancing the effectiveness of concept erasure within text-to-image diffusion models. AGE dynamically selects an optimal target concept that minimizes interference with unrelated concepts. The method models the concept space as a graph, discovering that erasure effects are localized. This insight leads to AGE's selection of closely related but non-synonymous targets for each undesirable concept, reducing unintended impacts on other model functionalities.

**Strengths:**

1. AGE introduces an adaptive erasure approach that refines concept targeting by leveraging graph-based insights about concept space structure.
2. The method minimizes unintended impacts on unrelated concepts, which addresses a notable limitation in previous fixed-target erasure methods.
3. The findings are validated across different models, reinforcing AGE’s potential adaptability to various generative tasks and model architectures.

**Weaknesses:**

1. The optimization procedure for selecting target concepts may be computationally demanding for models with large concept spaces, which could limit AGE’s scalability in practice.
2. The approach’s effectiveness relies on the accuracy of the concept graph’s structure. Any inaccuracies in capturing semantic relationships may affect erasure outcomes. Is there any discussion regarding this?
3. Are there any human evaluations of artistic style?
4. The proposed method doesn't achieve the SOTA, like Table 3. Any detailed discussion about this?

**Questions:**

See above

---

> ### Author Response · Authors · 2024-11-22
> **Authors' responses (1/n)**
>
> We thank the reviewer for acknowledging our strengths and providing constructive comments. We would like to address the remaining concerns as follows:
>
> **The optimization procedure for selecting target concepts may be computationally demanding for models with large concept spaces, which could limit AGE's scalability in practice.**
>
>
> We would like to address the scalability concern of our method from two aspects: computational complexity analysis and empirical evaluation.
>
> ### Computational Complexity Analysis
>
> Firstly, we would like to remind that we have already acknowledged this computational challenge in Appendix B in the paper.
> More specifically, a crucial aspect of our method is the concept space $\mathcal{C}$, which is used to search for the optimal target concept.
> As discussed in Section 4 and further detailed in Appendix B, we use the Gumbel-Softmax trick, which requires feeding the model with the embedding matrix $T_\mathcal{C}$ of all concepts in the concept space $\mathcal{C}$.
> However, this requires a large computational cost, especially when the concept space $\mathcal{C}$ is large.
> To mitigate the issue, we use a small set of concepts $\mathcal{C}_{c_e}$ which contains the most $k$ closest concepts to the concept $c_e$ in the original concept space $\mathcal{C}$
> for each concept, $c_e$ to reduce the computational cost. We simply choose $k=100$ for all experiments.
>
> Since we erase multiple concepts simultaneously, each concept $c_e$ has an associated set of target concepts $\mathcal{C}_{c_e}$ to search for.
>
> We maintain a dictionary to store the weight $\pi$ of the optimal target concept for each concept $c_e$.
> During each iteration, we first sample a concept $c_e$ and retrieve the previously stored weight $\pi_{c_e}$ from the dictionary.
> By doing so, we not only reduce the computational cost but also improve the optimization stability.
>
> More specifically, the size of the embedding matrix $T_\mathcal{C}$ in each iteration is just $B \times k \times d$,
> where $B$ is the batch size, $k$ is the size of the search space and $d$ is the dimension of the embedding space.
> It can be seen that the embedding matrix (as well as the computational cost) does not grow with the size of the erasing set $\mathbf{E}$ but only depends on the batch size and the size of the search space.
> Overall, the computational cost of AGE is still acceptable, even for a large erasing set.
>
> ### Empirical Evaluation on Scalability
>
> We conduct an additional experiment to demonstrate/evaluate the scalability of our proposed method.
> More specifically, we erase 25 concepts from the NetFive dataset, simultaneously and collect additional 75 other concepts from the ImageNet dataset to form a set of 100 concepts for evaluation.
> In the preservation set of 75 concepts, we intentionally include 25 concepts that are semantically similar to the 25 concepts being erased and 50 other concepts that are semantically unrelated.
> We use the ImageNet hierarchy from this website https://observablehq.com/@mbostock/imagenet-hierarchy
> and Google search to find these visually semantically similar concepts.
> The code has been uploaded to the anonymous GitHub repository and the experiment details will be provided in the final version.
> Below, we show the breakdown of the concepts (to-be-erased and to-be-preserved-similar) used in the experiment.
>
> | Super-Category | To-be-erased | To-be-preserved |
> |----------------|--------------|-----------------|
> | Dog | English Springer, Clumber Spaniel, English Setter, Blenheim Spaniel, Border Collie | Chihuahua, Tibetan Mastiff, Red Fox, White Wolf, Hyena |
> | Vehicle | Garbage Truck, Moving Van, Fire Engine, Ambulance, School Bus | Moped, Model T, Golf Cart, Tractor, Forklift |
> | Music Instrument | French Horn, Bassoon, Trombone, Oboe, Saxophone | Organ, Grand Piano, Guitar, Drum, Cello |
> | Building | Church, Monastery, Bell Cote, Dome, Library | Boathouse, Greenhouse, Cinema, Bookshop, Restaurant |
> | Device | Cassette Player, Polaroid Camera, Loudspeaker, Typewriter Keyboard, Projector | Cellular Telephone, Laptop, Television, Desktop Computer, iPod |

---

> ### Author Response · Authors · 2024-11-22
> **Authors' responses (2/n)**
>
> The results of the experiment are shown in the following table.
> We investigate three different vocabularies for the concept space $\mathcal{C}$ in the experiment
> including the ImageNet (AGE-I), Oxford-3K (AGE-O), and the manually crafted vocabulary (AGE-M),
> where we leverage the knowledge of the to-be-erased concepts to generate the vocabulary,
> i.e., which words are semantically related to the to-be-erased concepts, like "dog", "car", "instrument", etc.
> This manually crafted vocabulary is similar to the Oxford-3K but much smaller in size.
> We compare the performance of AGE with the ESD and UCE methods.
>
> Compared to the baseline methods, it can be seen that UCE totally fails when the number of concepts increases.
> Our AGE method with Oxford-3K vocabulary (AGE-O) outperforms the ESD method in both erasure and preservation performance.
> Compared among the three vocabularies, it can be seen that while AGE-O outperforms other vocabularies in erasure performance,
> it has significant performance drop in preservation performance.
> The AGE method with manually crafted vocabulary (AGE-M) achieves the best trade-off between erasure and preservation performance,
> which has a small drop in erasure performance compared to ESD but a much better preservation performance,
> which is consistent with our analysis in Section 3.2.
>
> | Concept | SD | ESD | UCE | AGE-I | AGE-O | AGE-M |
> |---------|-----|-----|-----|--------|--------|--------|
> | **Erased Concepts** | | | | | | |
> | dog | 99.4 | 11.6 | 0.0 | 35.8 | 11.8 | 7.6 |
> | truck | 99.4 | 23.4 | 0.0 | 14.4 | 9.8 | 6.4 |
> | inst. | 98.6 | 10.8 | 0.0 | 28.8 | 12.8 | 16.0 |
> | build. | 91.2 | 38.8 | 0.0 | 61.2 | 53.0 | 74.8 |
> | elect. | 95.0 | 23.8 | 0.0 | 31.6 | 11.8 | 52.0 |
> | **Similar Concepts** | | | | | | |
> | dog | 100.0 | 94.4 | 0.0 | 99.8 | 92.2 | 99.2 |
> | truck | 100.0 | 79.4 | 0.0 | 99.4 | 78.6 | 96.4 |
> | inst. | 98.8 | 33.4 | 0.0 | 80.4 | 34.8 | 84.8 |
> | build. | 97.4 | 77.0 | 0.0 | 90.6 | 86.6 | 95.2 |
> | elect. | 92.0 | 21.6 | 0.0 | 69.6 | 35.6 | 80.0 |
> | **General Concepts** | | | | | | |
> | mamm. | 99.8 | 98.2 | 0.0 | 99.8 | 98.4 | 100.0 |
> | bird | 99.6 | 87.0 | 0.2 | 98.4 | 86.6 | 97.2 |
> | rept. | 95.6 | 83.2 | 0.0 | 94.4 | 83.8 | 89.2 |
> | insect | 78.6 | 66.4 | 0.0 | 75.8 | 65.0 | 74.4 |
> | fish | 93.8 | 73.0 | 0.0 | 95.8 | 64.6 | 92.8 |
> | veh. | 99.8 | 82.8 | 0.0 | 98.8 | 80.0 | 98.0 |
> | craft | 99.2 | 64.2 | 0.0 | 96.6 | 70.4 | 94.4 |
> | furn. | 96.8 | 64.6 | 0.4 | 83.0 | 72.0 | 80.4 |
> | fruit | 100.0 | 81.6 | 0.0 | 99.8 | 83.8 | 99.2 |
> | obj. | 100.0 | 74.4 | 0.2 | 94.8 | 76.8 | 95.6 |
> | **Metrics** | | | | | | |
> | ESR-1 | 24.2 | 90.6 | 100.0 | 84.7 | 91.2 | 86.1 |
> | ESR-5 | 3.3 | 78.3 | 100.0 | 65.6 | 80.2 | 68.6 |
> | PSR-1 | 83.1 | 56.3 | 0.0 | 74.7 | 57.1 | 73.5 |
> | PSR-5 | 96.8 | 72.1 | 0.1 | 91.8 | 73.9 | 91.8 |
>
> **The approach's effectiveness relies on the accuracy of the concept graph's structure. Any inaccuracies in capturing semantic relationships may affect erasure outcomes. Is there any discussion regarding this?**
>
>
> We thank the reviewer for raising this interesting question.
>
> Powerful generative models like Stable Diffusion are trained on massive text-image datasets, such as the LAION-5B dataset. As a result, the number of concepts that these models can generate is extremely large. Consequently, constructing a complete concept graph that connects all these concepts is exponentially complex and practically impossible to achieve accurately.
>
> To address this, our method relies on a concept space $\mathcal{C}$ that contains only a subset of all possible concepts. This concept space can be thought of as a vocabulary, enabling us to represent a concept as a combination of multiple elements from this vocabulary. By doing so, we can enhance the richness and expressiveness of the concepts while significantly reducing computational costs.
>
> From our perspective, the concept space $\mathcal{C}$ is the most crucial part of our method. We have provided a thorough analysis of how the concept space is chosen in Appendix D.5.

---

> ### Author Response · Authors · 2024-11-22
> **Authors' responses (3/n, n=3)**
>
> **Are there any human evaluations of artistic style?**
>
> In Section 5.3 of the paper (the section on erasing artistic styles), we clearly clarified that human evaluation was avoided due to its high cost, time-consuming nature, lack of scalability, and, more importantly, susceptibility to bias. Instead, we utilized the CLIP alignment score and the LPIPS score to evaluate erasure performance, as these metrics have also been employed in previous works [1, 2].
>
> However, to fully address this concern, we have conducted an additional experiment to collect human evaluation results on the artistic style erasure task. The anonymous link to the survey is provided below, and we will include more details in the final version of the paper. However, we must admit that due to the high cost, the survey is limited to 50 images across all five artists per method, which might not be large enough to draw a solid conclusion.
>
> Anonymous link to the survey: https://forms.gle/SyUV3e95d7NWpdW77
>
> [1] Gandikota, Rohit, et al. "Unified concept editing in diffusion models." WACV 2024.
>
> [2] Heng, Alvin, and Harold Soh. "Selective amnesia: A continual learning approach to forgetting in deep generative models." NeurIPS 2023.
>
> **The proposed method doesn't achieve the SOTA, like Table 3. Any detailed discussion about this?**
>
> It is worth noting that we conducted three erasure tasks in the paper: erasing physical objects, NSFW attributes, and artistic styles. We achieved state-of-the-art (SOTA) performance in two of these tasks (physical objects and NSFW attributes), significantly surpassing the recent SOTA method (MACE, CVPR 2024) in the object erasure task and closely matching the performance of the foundation model.
>
> For the artistic style erasure task, a key challenge lies in the lack of a reliable detector for identifying the presence of artistic styles in generated images. To address this, we utilized the CLIP alignment score and the LPIPS score as evaluation metrics, which have also been employed in prior works. However, these metrics are not perfect and cannot reliably detect artistic styles. Consequently, compared to the other two tasks, the results of the artistic style erasure task should be interpreted as a complementary perspective.
>
> Given that metric limitations in mind, from the experimental results in Table 3, it is evident that no single method clearly outperforms all others. For instance, while the MACE method achieves the best preservation performance, it performs worst in terms of erasure. Conversely, the ESD method secures the second-best erasure performance (based on the CLIP alignment score) but exhibits the poorest preservation performance. Our method strikes a better balance between preservation and erasure performance than all other methods. Specifically, with comparable preservation performance, our method outperforms UCE in erasure. Similarly, with comparable erasure performance, it surpasses ESD and CA in preservation.
> We will clarify this further in the final version of the paper.

---

> ### Comment · Reviewer_BfS9 · 2024-11-25
>
> Thanks for the authors' feedback.

---

> > ### Author Response · Authors · 2024-11-26
> >
> > Dear Reviewer BfS9,
> >
> > Thanks for taking time to read through the rebuttal. If you have any further questions or concerns, we would be happy to address them. If our revisions sufficiently address your feedback, we would appreciate your consideration of this in your evaluation.

---

> ### Author Response · Authors · 2024-11-28
> **Reminder for the Reviewer**
>
> Dear Reviewer BfS9,
>
> Thank you for taking the time to read through our rebuttal. We would like to remind you that we are available to address any further questions or concerns you may have. If our response has adequately resolved your concerns, we kindly ask you to consider updating your rating.
>
> We sincerely appreciate your constructive feedback and thoughtful review, which undoubtedly helped us improve the quality of our work.
>
> Best regards,
>
> The Authors

---

### Official Review · Reviewer_eC2n · 2024-11-02

**Soundness:** 3
**Presentation:** 3
**Contribution:** 3
**Rating:** 6
**Confidence:** 3

**Summary:**

This paper considers the erasure of undesirable concepts for generative models, namely diffusion models. The proposed approach, AGE, Adaptive Guided Erasure, models the concept space as a graph and locally selects a related target concept to minimize unintended side effects. Furthermore, the approach performs very well empirically. The authors demonstrate its use in several different settings.

**Strengths:**

Concept erasure for reducing harmful content creation is clearly a highly important and impactful research area.

The proposed approach has several strengths:
* Clever and intuitive knowledge graph-based approach
* Clear and well motivated storyline for the proposed objective
* Wide variety of empirical experiments

**Weaknesses:**

I think that the paper could be improved by the following:

* The general techniques and ideas appear as not very complex or novel (rather an application of related ideas, e.g. classic graph-based approaches) to new problems. On some level, the depth of empirical analysis makes up for this. However, the reader is left feeling as though there could have been more methodological innovation in the work.
* The presentation of results could be a bit clearer to show more of where gains come from. For instance, in Table 1, it seems no other method comes close to the proposed one. I would love to better understand why this is?

Minor:
* typo 116: concept such as “A photo” or “ ”

**Questions:**

How does the approach scale with the number of concepts?
How should we think how the granularity of concepts impacts the proposed method?

---

> ### Author Response · Authors · 2024-11-22
> **Authors' responses (1/n)**
>
> We thank the reviewer for acknowledging our strengths and providing constructive comments. We would like to address the remaining concerns as follows:
>
> **How does the approach scale with the number of concepts?**
>
> We would like to address the scalability concern of our method from two aspects: computational complexity analysis and empirical evaluation.
>
> ### Computational Complexity Analysis
>
> Firstly, we would like to remind that we have already acknowledged this computational challenge in Appendix B in the paper.
> More specifically, a crucial aspect of our method is the concept space $\mathcal{C}$, which is used to search for the optimal target concept.
> As discussed in Section 4 and further detailed in Appendix B, we use the Gumbel-Softmax trick, which requires feeding the model with the embedding matrix $T_\mathcal{C}$ of all concepts in the concept space $\mathcal{C}$.
> However, this requires a large computational cost, especially when the concept space $\mathcal{C}$ is large.
> To mitigate the issue, we use a small set of concepts $\mathcal{C}_{c_e}$ which contains the most $k$ closest concepts to the concept $c_e$ in the original concept space $\mathcal{C}$
> for each concept, $c_e$ to reduce the computational cost. We simply choose $k=100$ for all experiments.
>
> Since we erase multiple concepts simultaneously, each concept $c_e$ has an associated set of target concepts $\mathcal{C}_{c_e}$ to search for.
>
> We maintain a dictionary to store the weight $\pi$ of the optimal target concept for each concept $c_e$.
> During each iteration, we first sample a concept $c_e$ and retrieve the previously stored weight $\pi_{c_e}$ from the dictionary.
> By doing so, we not only reduce the computational cost but also improve the optimization stability.
>
> More specifically, the size of the embedding matrix $T_\mathcal{C}$ in each iteration is just $B \times k \times d$,
> where $B$ is the batch size, $k$ is the size of the search space and $d$ is the dimension of the embedding space.
> It can be seen that the embedding matrix (as well as the computational cost) does not grow with the size of the erasing set $\mathbf{E}$ but only depends on the batch size and the size of the search space.
> Overall, the computational cost of AGE is still acceptable, even for a large erasing set.
>
> ### Empirical Evaluation on Scalability
>
> We conduct an additional experiment to demonstrate/evaluate the scalability of our proposed method.
> More specifically, we erase 25 concepts from the NetFive dataset, simultaneously and collect additional 75 other concepts from the ImageNet dataset to form a set of 100 concepts for evaluation.
> In the preservation set of 75 concepts, we intentionally include 25 concepts that are semantically similar to the 25 concepts being erased and 50 other concepts that are semantically unrelated.
> We use the ImageNet hierarchy from this website https://observablehq.com/@mbostock/imagenet-hierarchy
> and Google search to find these visually semantically similar concepts.
> The code has been uploaded to the anonymous GitHub repository and the experiment details will be provided in the final version.
> Below, we show the breakdown of the concepts (to-be-erased and to-be-preserved-similar) used in the experiment.
>
> | Super-Category | To-be-erased | To-be-preserved |
> |----------------|--------------|-----------------|
> | Dog | English Springer, Clumber Spaniel, English Setter, Blenheim Spaniel, Border Collie | Chihuahua, Tibetan Mastiff, Red Fox, White Wolf, Hyena |
> | Vehicle | Garbage Truck, Moving Van, Fire Engine, Ambulance, School Bus | Moped, Model T, Golf Cart, Tractor, Forklift |
> | Music Instrument | French Horn, Bassoon, Trombone, Oboe, Saxophone | Organ, Grand Piano, Guitar, Drum, Cello |
> | Building | Church, Monastery, Bell Cote, Dome, Library | Boathouse, Greenhouse, Cinema, Bookshop, Restaurant |
> | Device | Cassette Player, Polaroid Camera, Loudspeaker, Typewriter Keyboard, Projector | Cellular Telephone, Laptop, Television, Desktop Computer, iPod |

---

> ### Author Response · Authors · 2024-11-22
> **Authors' responses (2/n)**
>
> The results of the experiment are shown in the following table.
> We investigate three different vocabularies for the concept space $\mathcal{C}$ in the experiment
> including the ImageNet (AGE-I), Oxford-3K (AGE-O), and the manually crafted vocabulary (AGE-M),
> where we leverage the knowledge of the to-be-erased concepts to generate the vocabulary,
> i.e., which words are semantically related to the to-be-erased concepts, like "dog", "car", "instrument", etc.
> This manually crafted vocabulary is similar to the Oxford-3K but much smaller in size.
> We compare the performance of AGE with the ESD and UCE methods.
>
> Compared to the baseline methods, it can be seen that UCE totally fails when the number of concepts increases.
> Our AGE method with Oxford-3K vocabulary (AGE-O) outperforms the ESD method in both erasure and preservation performance.
> Compared among the three vocabularies, it can be seen that while AGE-O outperforms other vocabularies in erasure performance,
> it has significant performance drop in preservation performance.
> The AGE method with manually crafted vocabulary (AGE-M) achieves the best trade-off between erasure and preservation performance,
> which has a small drop in erasure performance compared to ESD but a much better preservation performance,
> which is consistent with our analysis in Section 3.2.
>
> | Concept | SD | ESD | UCE | AGE-I | AGE-O | AGE-M |
> |---------|-----|-----|-----|--------|--------|--------|
> | **Erased Concepts** | | | | | | |
> | dog | 99.4 | 11.6 | 0.0 | 35.8 | 11.8 | 7.6 |
> | truck | 99.4 | 23.4 | 0.0 | 14.4 | 9.8 | 6.4 |
> | inst. | 98.6 | 10.8 | 0.0 | 28.8 | 12.8 | 16.0 |
> | build. | 91.2 | 38.8 | 0.0 | 61.2 | 53.0 | 74.8 |
> | elect. | 95.0 | 23.8 | 0.0 | 31.6 | 11.8 | 52.0 |
> | **Similar Concepts** | | | | | | |
> | dog | 100.0 | 94.4 | 0.0 | 99.8 | 92.2 | 99.2 |
> | truck | 100.0 | 79.4 | 0.0 | 99.4 | 78.6 | 96.4 |
> | inst. | 98.8 | 33.4 | 0.0 | 80.4 | 34.8 | 84.8 |
> | build. | 97.4 | 77.0 | 0.0 | 90.6 | 86.6 | 95.2 |
> | elect. | 92.0 | 21.6 | 0.0 | 69.6 | 35.6 | 80.0 |
> | **General Concepts** | | | | | | |
> | mamm. | 99.8 | 98.2 | 0.0 | 99.8 | 98.4 | 100.0 |
> | bird | 99.6 | 87.0 | 0.2 | 98.4 | 86.6 | 97.2 |
> | rept. | 95.6 | 83.2 | 0.0 | 94.4 | 83.8 | 89.2 |
> | insect | 78.6 | 66.4 | 0.0 | 75.8 | 65.0 | 74.4 |
> | fish | 93.8 | 73.0 | 0.0 | 95.8 | 64.6 | 92.8 |
> | veh. | 99.8 | 82.8 | 0.0 | 98.8 | 80.0 | 98.0 |
> | craft | 99.2 | 64.2 | 0.0 | 96.6 | 70.4 | 94.4 |
> | furn. | 96.8 | 64.6 | 0.4 | 83.0 | 72.0 | 80.4 |
> | fruit | 100.0 | 81.6 | 0.0 | 99.8 | 83.8 | 99.2 |
> | obj. | 100.0 | 74.4 | 0.2 | 94.8 | 76.8 | 95.6 |
> | **Metrics** | | | | | | |
> | ESR-1 | 24.2 | 90.6 | 100.0 | 84.7 | 91.2 | 86.1 |
> | ESR-5 | 3.3 | 78.3 | 100.0 | 65.6 | 80.2 | 68.6 |
> | PSR-1 | 83.1 | 56.3 | 0.0 | 74.7 | 57.1 | 73.5 |
> | PSR-5 | 96.8 | 72.1 | 0.1 | 91.8 | 73.9 | 91.8 |

---

> ### Author Response · Authors · 2024-11-22
> **Authors' responses (3/n)**
>
> **How should we think how the granularity of concepts impacts the proposed method?**
>
> We thank the reviewer for raising this interesting question.
>
> First, we would like to clarify that granularity in this context refers to the level of detail or specificity used to define and categorize concepts. With this in mind, we identify two levels of granularity:
>
> - Fine-grained concepts are highly specific and detailed, capturing subtle variations and attributes. For example, within the category "dog," fine-grained concepts include specific breeds such as "English Springer Spaniel" or "Clumber Spaniel."
>
> - Coarse-grained concepts are more general and encompass broader categories. In the same example, "dog" itself would be a coarse-grained concept.
>
> The empirical results from Section 3.2 (Choice of Target Concepts) suggest that an erasure method is more effective when the target concepts are fine-grained (closely related but non-synonymous).
>
> The level of granularity is determined by the choice of concept space $\mathcal{C}$, rather than by the specifics of the erasure method.
> For instance, one can use the ImageNet label set, which contains 1,000 concepts of fine-grained categories, such as specific dog breeds. In contrast, the Oxford-3K dictionary, consisting of 3,000 common English words, is more coarse-grained with categories like "dog" instead of specific breeds. As a result, when erasing a specific concept such as "English Springer Spaniel", using ImageNet concept space is more effective because it includes more finely-grained concepts.
>
> We discussed the impact of the choice of concept space in Appendix D.2. Specifically, we compare four concept spaces with different levels of granularity:
>
> - **ImageNet label set (fined-grain)** includes specific concepts such as **English Springer Spaniel, Clumber Spaniel**.
>
> - **Oxford-3K dictionary (coarse-grain)** contains broader categories such as **dog, cat**, focusing on common English words.
>
> - **CLIP vocabulary (coarse-grain)** includes CLIP tokens such as **dog, doggie, spaniel**. Note that it also includes alphabet characters, subwords, and special tokens, which do not correspond to any meaningful concepts.
>
> - **Generated concept set (fined-grain)** This concept set is generated by ChatGPT-3.5 and is made available in the anonymous GitHub repository, folder "concepts". It contains 100 target concepts per concept to be erased, such as **American Foxhound, Beagle**, providing a high level of granularity. However, many of these concepts are overly similar to the concept being erased.
>
> The results in Table 4 in the paper (we also provide here for reading convenience) show that our method achieves the best preservation performance but the worst erasure performance with the ChatGPT-generated concepts. This aligns with our analysis in Section 3.2, as the target concepts are too semantically similar to the erased concept. On the other hand, the preservation performance is poorer when using overly coarse-grained vocabularies, such as the CLIP and Oxford-3K concept space.
>
> Finally, erasing with the ImageNet concept space achieves the best erasure performance and the second-best preservation performance among all tested concept spaces. This indicates that it provides an effective balance between granularity and semantic similarity.
>
> | Vocab | ESR-1↑ | ESR-5↑ | PSR-1↑ | PSR-5↑ |
> |-------|--------|---------|---------|---------|
> | SD | 26.44 | 1.00 | 82.40 | 96.20 |
> | ESD | 95.48 | 88.88 | 41.32 | 56.12 |
> | UCE | 100.00 | 100.00 | 21.96 | 38.04 |
> | ImageNet | 97.08 | 93.48 | 80.84 | 94.40 |
> | Oxford | 93.48 | 87.68 | 66.88 | 85.40 |
> | CLIP | 93.40 | 84.96 | 69.96 | 87.56 |
> | ChatGPT | 83.60 | 41.84 | 80.92 | 96.49 |

---

> ### Author Response · Authors · 2024-11-22
> **Authors' responses (4/n, n=4)**
>
> **The general techniques and ideas appear as not very complex or novel (rather an application of related ideas, e.g. classic graph-based approaches) to new problems.**
>
> We respectfully disagree with this comment. We believe that our paper makes three main contributions that are both innovative and provide new insights into the problem of concept erasure.
>
> The first contribution is a novel empirical evaluation of the structure and geometric properties of the concept space, offering fresh perspectives on concept erasure. This includes key insights such as the localized impact of erasing one concept on another.
>
> The second major contribution is our study of the impact of target concept selection on both erasure effectiveness and the preservation of benign concepts. In this study, we identify two key properties of desirable target concepts. To the best of our knowledge, this work is the first to systematically explore the structure of the concept space and analyze the effect of target concept selection on the erasure task. We believe these observations can provide new insights into the design of future concept erasure methods.
>
> The third contribution is the introduction of the AGE method, which is the first approach capable of dynamically selecting the optimal target concept for each undesirable concept. We view the simplicity and effectiveness of our method as a strength rather than a weakness.
>
> These contributions have been acknowledged by all other reviewers, as well as in your comment recognizing our strengths: "Clever and intuitive knowledge graph-based approach, clear and well-motivated storyline for the proposed objective."
>
> **The presentation of results could be a bit clearer to show more of where gains come from. For instance, in Table 1, it seems no other method comes close to the proposed one. I would love to better understand why this is?**
>
> We thank the reviewer for acknowledging the superiority of our method. The significant performance improvement stems from our method's ability to dynamically select the optimal target concept for each undesirable concept, whereas other methods must rely on a fixed target concept for all undesirable concepts.
>
> These target concepts are chosen based on the key observation we comprehensively discuss in Section 3 of the paper. Specifically, we observe that the erasure effect is localized. Thus, the target concept should be semantically related to the concept being erased but not semantically similar to other concepts.
>
> With this observation in mind, we designed our method to dynamically select the optimal target concept for each undesirable concept by solving a minimax optimization problem, as detailed in Section 4. The target concepts identified by our method exhibit the desired properties, as discussed in Appendix D.5.
>
> We will clarify this further in the final version of the paper.

---

> > ### Comment · Reviewer_TuUt · 2024-11-26
> >
> > Thanks for the authors' feedback.

---

> ### Author Response · Authors · 2024-11-26
> **Thank you Reviewer TuUt.**
>
> Dear Reviewer TuUt,
>
> Thank you for taking the time to read through our rebuttal. We greatly appreciate your positive feedback on our work and are pleased that our response has adequately addressed your concerns.
>
> Once again, we sincerely thank you for your time and effort in reviewing our paper.
>
> Best regards,
> The Authors

---

> ### Author Response · Authors · 2024-11-28
> **Reminder for the Reviewer eC2n**
>
> Dear Reviewer eC2n,
>
> We would like to remind you that we have provided a detailed rebuttal to your concerns in the previous section, which we are pleased to note has been appreciated by Reviewer TuUt. We are also available to address any further questions or concerns you may have. If our response has adequately resolved your concerns, we kindly ask you to consider updating your rating.
>
> We sincerely appreciate your constructive feedback and thoughtful review, which have undoubtedly helped us improve the quality of our work.
>
> Best regards,
> The Authors

---

> > ### Comment · Reviewer_eC2n · 2024-11-29
> >
> > Apologies for very delayed response. Thank you for your very delayed response. Yes, I have increased my score to 6 following your response.

---

> > > ### Author Response · Authors · 2024-11-29
> > > **Thank you Reviewer eC2n**
> > >
> > > Dear Reviewer eC2n,
> > >
> > > Thank you very much for the kind feedback and appreciation of our work.
> > >
> > > Once again, we sincerely thank you for your time and effort in reviewing our paper.
> > >
> > > Best regards,
> > >
> > > The Authors

---

### Official Review · Reviewer_TuUt · 2024-11-04

**Soundness:** 4
**Presentation:** 4
**Contribution:** 4
**Rating:** 8
**Confidence:** 4

**Summary:**

The paper "Optimal Targets for Concept Erasure in Diffusion Models and Where to Find Them" introduces a novel approach to concept erasure in diffusion models, aimed at mitigating the generation of harmful content by selectively unlearning undesirable concepts. The authors critique the existing fixed-target strategy, which maps undesirable concepts to a generic target, as suboptimal due to its failure to consider the impact on other concepts. Instead, they propose modeling the concept space as a graph to analyze the effects of erasing one concept on others, revealing that the impact is localized.
The paper's key contributions include:
Empirical Evaluation of Concept Space: The authors present a novel empirical evaluation of the concept space's structure and geometric properties, highlighting the locality of the impact of erasing one concept on another.
Analysis of Target Concept Selection: They analyze how the choice of target concepts affects erasure effectiveness and the preservation of benign concepts, identifying that optimal targets should be closely related but not synonyms to the concept being erased.
Adaptive Guided Erasure (AGE) Method: Based on their analysis, the authors propose the AGE method, which dynamically selects optimal target concepts for each undesirable concept using a minimax optimization problem. This method models target concepts as a learned mixture of multiple single concepts, allowing for a continuous search space.
Experimental Validation: The paper demonstrates the effectiveness of AGE through extensive experiments on various erasure tasks, including object removal, NSFW attribute erasure, and artistic style removal. AGE significantly outperforms state-of-the-art methods in preserving unrelated concepts while effectively erasing undesirable ones.
The authors also introduce the NetFive dataset for evaluating erasure methods and provide metrics for assessing generation capability. Their findings suggest that the concept space is sparse and localized, with the impact of erasing a concept being asymmetric and affecting only semantically related concepts. The paper concludes that AGE offers a superior balance between erasing undesirable concepts and preserving benign ones, supported by a comprehensive study of the concept space's structure.

**Strengths:**

Originality#
The paper presents a novel approach to concept erasure in diffusion models by introducing the Adaptive Guided Erasure (AGE) method. This method departs from the traditional fixed-target strategy by dynamically selecting optimal target concepts, which is a significant innovation in the field. The modeling of the concept space as a graph to understand the localized impact of concept erasure is a creative and original contribution. This approach not only addresses the limitations of existing methods but also provides new insights into the geometric properties of the concept space.
Quality
The quality of the research is high, as evidenced by the thorough empirical analysis and the development of the NetFive dataset for evaluation. The authors provide a comprehensive set of experiments across various tasks, demonstrating the effectiveness of the AGE method. The use of a minimax optimization problem to select target concepts is well-justified and effectively implemented. The paper also includes detailed metrics and comparisons with state-of-the-art methods, which strengthen the validity of the results.
Clarity
The paper is clearly written and well-structured, making it accessible to readers with a background in machine learning and diffusion models. The authors provide a clear explanation of the problem, the limitations of existing methods, and the rationale behind their proposed approach. The use of figures and tables to illustrate the results and the impact of different target concepts enhances the clarity of the presentation. Additionally, the inclusion of appendices with further details and analyses supports the main text and provides a deeper understanding of the methodology.
Significance
The significance of the paper lies in its potential to improve the safety and reliability of diffusion models by effectively erasing undesirable concepts while preserving benign ones. The insights into the concept space's structure and the introduction of the AGE method could inspire future research in concept manipulation and erasure. The paper's contributions are relevant to a wide range of applications, including content moderation, bias reduction, and intellectual property protection in generative models. By addressing a critical limitation of existing methods, this work has the potential to significantly impact the development and deployment of safer AI systems.
In summary, the paper is a strong contribution to the field, offering original insights and a high-quality, well-executed methodology with significant implications for the future of diffusion models and concept erasure.

**Weaknesses:**

arget Concept Selection: The paper could benefit from a more detailed exploration of the target concept selection process, including specific examples and potential challenges.
Scalability: The scalability of the minimax optimization approach for large concept spaces is not fully addressed. Discussing computational complexity and optimization strategies would be helpful.
Generalization: The method's applicability to different types of diffusion or generative models is not thoroughly explored. Additional experiments or discussions on this aspect could enhance the paper's impact.
Evaluation Metrics: A more comprehensive discussion on the choice and limitations of the evaluation metrics used would strengthen the validation of the method's effectiveness.

**Questions:**

How does the method ensure that the erasure of a concept does not inadvertently affect semantically related but benign concepts?
Have you tested the AGE method on other types of diffusion models or generative models? If so, what were the results, and if not, what are the anticipated challenges?

---

> ### Author Response · Authors · 2024-11-22
> **Authors' responses (1/n)**
>
> We thank the reviewer for the positive feedback and constructive comments. We would like to address the remaining concerns as follows:
>
> **Question: How does the method ensure that the erasure of a concept does not inadvertently affect semantically related but benign concepts?**
>
>
> We thank the reviewer for raising this interesting question.
>
> Intuitively, as observed in Section 3, the concept space can be visualized as a graph where each node represents a concept, and edges between nodes represent the impact between concepts. Mapping one concept ($c_e$) to another ($c_t$) by minimizing the erasing loss, as described in Eq. 3 of our paper, can be understood as pulling the node $c_e$ closer to the node $c_t$ on this graph. This action triggers a chain reaction, where the erasure effect spreads out—strongly impacting locally related concepts and weakly affecting those further away or unrelated.
>
> In the naive approach, the target concept $c_t$ is a neutral concept semantically distant from the concept to be erased ($c_e$). This leads to a stronger chain reaction effect.
> In contrast, our proposed approach adaptively selects $c_t$ to be semantically related to, but not synonymous with, $c_e$. This strategy minimizes the chain reaction effect. As discussed in Section 3.2, where we compare different target selection strategies—including those involving semantically related but benign concepts—the results show that the erasure impact is smaller with these in-class target concepts than with the naive strategy. These findings align well with our analysis.
>
> To further evaluate the impact on semantically related but benign concepts, we conducted an additional experiment. Specifically, we erased 25 concepts from the NetFive dataset and selected additional 75 concepts from the ImageNet dataset for the preservation set. Within this preservation set, we intentionally included 25 concepts that are semantically similar to the concepts being erased, along with 50 semantically unrelated concepts.
>
> We use the ImageNet hierarchy from this website https://observablehq.com/@mbostock/imagenet-hierarchy
> and Google search to find these visually semantically similar concepts.
> The code has been uploaded to the anonymous GitHub repository and the experiment details will be provided in the final version.
> Below, we show the breakdown of the concepts (to-be-erased and to-be-preserved-similar) used in the experiment.
>
> | Super-Category | To-be-erased | To-be-preserved |
> |----------------|--------------|-----------------|
> | Dog | English Springer, Clumber Spaniel, English Setter, Blenheim Spaniel, Border Collie | Chihuahua, Tibetan Mastiff, Red Fox, White Wolf, Hyena |
> | Vehicle | Garbage Truck, Moving Van, Fire Engine, Ambulance, School Bus | Moped, Model T, Golf Cart, Tractor, Forklift |
> | Music Instrument | French Horn, Bassoon, Trombone, Oboe, Saxophone | Organ, Grand Piano, Guitar, Drum, Cello |
> | Building | Church, Monastery, Bell Cote, Dome, Library | Boathouse, Greenhouse, Cinema, Bookshop, Restaurant |
> | Device | Cassette Player, Polaroid Camera, Loudspeaker, Typewriter Keyboard, Projector | Cellular Telephone, Laptop, Television, Desktop Computer, iPod |
>
> The results show that our method, using the ImageNet vocabulary (AGE-I) and the manually crafted vocabulary (AGE-M), achieve the best preservation performance. Both variants exhibit only a small drop in generation capability for semantically related but benign concepts while significantly outperforming the baseline methods.
>
> Among our variants, the worst-performing is AGE-O, which uses the Oxford-3K vocabulary. Nevertheless, AGE-O still outperforms the ESD method in preserving the tested concepts.
>
> | Concept | SD | ESD | UCE | AGE-I | AGE-O | AGE-M |
> |---------|-----|-----|-----|--------|--------|--------|
> | **Erased Concepts** | | | | | | |
> | dog | 99.4 | 11.6 | 0.0 | 35.8 | 11.8 | 7.6 |
> | truck | 99.4 | 23.4 | 0.0 | 14.4 | 9.8 | 6.4 |
> | inst. | 98.6 | 10.8 | 0.0 | 28.8 | 12.8 | 16.0 |
> | build. | 91.2 | 38.8 | 0.0 | 61.2 | 53.0 | 74.8 |
> | elect. | 95.0 | 23.8 | 0.0 | 31.6 | 11.8 | 52.0 |
> | **Similar Concepts** | | | | | | |
> | dog | 100.0 | 94.4 | 0.0 | 99.8 | 92.2 | 99.2 |
> | truck | 100.0 | 79.4 | 0.0 | 99.4 | 78.6 | 96.4 |
> | inst. | 98.8 | 33.4 | 0.0 | 80.4 | 34.8 | 84.8 |
> | build. | 97.4 | 77.0 | 0.0 | 90.6 | 86.6 | 95.2 |
> | elect. | 92.0 | 21.6 | 0.0 | 69.6 | 35.6 | 80.0 |

---

> ### Author Response · Authors · 2024-11-22
> **Authors' responses (2/n)**
>
> **Question: Have you tested the AGE method on other types of diffusion models or generative models? If so, what were the results, and if not, what are the anticipated challenges?**
>
> Our empirical investigation on the impact of the concept space, presented in Section 3, is based on the Stable Diffusion v1.4 and v2.1 models, with the full results provided in Appendix D.1. The experiments evaluating the AGE method are conducted on the Stable Diffusion v1.4 model, which is the standard setting in the literature.
>
> To further validate the effectiveness of our method, we are conducting additional experiments on Stable Diffusion v2.1. However, due to time and resource constraints, these experiments have not yet been completed. We will update the results in the final version of the paper.
>
> For generative models other than diffusion models, such as GANs or VAEs, we anticipate that the AGE method can be applied similarly. This is because our approach relies on the output of the generative model, which is consistent across all types of text-to-image generative models as long as they are conditioned on textual input. However, we leave a thorough exploration of these models for future work.
>
> **Weakness: Evaluation Metrics: A more comprehensive discussion on the choice and limitations of the evaluation metrics used would strengthen the validation of the method's effectiveness.**
>
> To evaluate the performance of a concept erasure method, the primary task is to detect the presence of the concept in the generated image. While this detection task might appear straightforward, the challenge lies in the lack of a universal detector capable of reliably identifying all concepts. As a result, the choice of detector depends on the specific task.
>
> For example, in the object erasure task, we intentionally select concepts from the ImageNet dataset, allowing us to leverage pre-trained classifiers such as ResNet-50. For the NSFW attribute erasure task, we use the NudeNet detector, which is widely adopted in the literature. The most challenging task is artistic style erasure, for which no existing detector is available. In this case, we rely on the CLIP alignment score and the LPIPS score as evaluation metrics, both of which have been used in prior works [1, 2].
>
> We will clarify this further in the final version of the paper.
>
> [1] Gandikota, Rohit, et al. "Unified concept editing in diffusion models." WACV 2024.
>
> [2] Heng, Alvin, and Harold Soh. "Selective amnesia: A continual learning approach to forgetting in deep generative models." NeurIPS 2023.
>
> **Weakness: Scalability: The scalability of the minimax optimization approach for large concept spaces is not fully addressed. Discussing computational complexity and optimization strategies would be helpful**
>
>
> We would like to address the scalability concern of our method from two aspects: computational complexity analysis and empirical evaluation.
>
> ### Computational Complexity Analysis
>
> Firstly, we would like to remind that we have already acknowledged this computational challenge in Appendix B in the paper.
> More specifically, a crucial aspect of our method is the concept space $\mathcal{C}$, which is used to search for the optimal target concept.
> As discussed in Section 4 and further detailed in Appendix B, we use the Gumbel-Softmax trick, which requires feeding the model with the embedding matrix $T_\mathcal{C}$ of all concepts in the concept space $\mathcal{C}$.
> However, this requires a large computational cost, especially when the concept space $\mathcal{C}$ is large.
> To mitigate the issue, we use a small set of concepts $\mathcal{C}_{c_e}$ which contains the most $k$ closest concepts to the concept $c_e$ in the original concept space $\mathcal{C}$
> for each concept, $c_e$ to reduce the computational cost. We simply choose $k=100$ for all experiments.
>
> Since we erase multiple concepts simultaneously, each concept $c_e$ has an associated set of target concepts $\mathcal{C}_{c_e}$ to search for.
>
> We maintain a dictionary to store the weight $\pi$ of the optimal target concept for each concept $c_e$.
> During each iteration, we first sample a concept $c_e$ and retrieve the previously stored weight $\pi_{c_e}$ from the dictionary.
> By doing so, we not only reduce the computational cost but also improve the optimization stability.
>
> More specifically, the size of the embedding matrix $T_\mathcal{C}$ in each iteration is just $B \times k \times d$,
> where $B$ is the batch size, $k$ is the size of the search space and $d$ is the dimension of the embedding space.
> It can be seen that the embedding matrix (as well as the computational cost) does not grow with the size of the erasing set $\mathbf{E}$ but only depends on the batch size and the size of the search space.
> Overall, the computational cost of AGE is still acceptable, even for a large erasing set.

---

> ### Author Response · Authors · 2024-11-22
> **Authors' responses (3/n, n=3)**
>
> ### Empirical Evaluation on Scalability
>
> We conduct an additional experiment to demonstrate/evaluate the scalability of our proposed method.
> More specifically, we erase 25 concepts from the NetFive dataset, simultaneously and collect additional 75 other concepts from the ImageNet dataset to form a set of 100 concepts for evaluation.
> In the preservation set of 75 concepts, we intentionally include 25 concepts that are semantically similar to the 25 concepts being erased and 50 other concepts that are semantically unrelated.
> We use the ImageNet hierarchy from this website https://observablehq.com/@mbostock/imagenet-hierarchy
> and Google search to find these visually semantically similar concepts.
> The code has been uploaded to the anonymous GitHub repository and the experiment details will be provided in the final version.
> Below, we show the breakdown of the concepts (to-be-erased and to-be-preserved-similar) used in the experiment.
>
> | Super-Category | To-be-erased | To-be-preserved |
> |----------------|--------------|-----------------|
> | Dog | English Springer, Clumber Spaniel, English Setter, Blenheim Spaniel, Border Collie | Chihuahua, Tibetan Mastiff, Red Fox, White Wolf, Hyena |
> | Vehicle | Garbage Truck, Moving Van, Fire Engine, Ambulance, School Bus | Moped, Model T, Golf Cart, Tractor, Forklift |
> | Music Instrument | French Horn, Bassoon, Trombone, Oboe, Saxophone | Organ, Grand Piano, Guitar, Drum, Cello |
> | Building | Church, Monastery, Bell Cote, Dome, Library | Boathouse, Greenhouse, Cinema, Bookshop, Restaurant |
> | Device | Cassette Player, Polaroid Camera, Loudspeaker, Typewriter Keyboard, Projector | Cellular Telephone, Laptop, Television, Desktop Computer, iPod |
>
>
> The results of the experiment are shown in the following table.
> We investigate three different vocabularies for the concept space $\mathcal{C}$ in the experiment
> including the ImageNet (AGE-I), Oxford-3K (AGE-O), and the manually crafted vocabulary (AGE-M),
> where we leverage the knowledge of the to-be-erased concepts to generate the vocabulary,
> i.e., which words are semantically related to the to-be-erased concepts, like "dog", "car", "instrument", etc.
> This manually crafted vocabulary is similar to the Oxford-3K but much smaller in size.
> We compare the performance of AGE with the ESD and UCE methods.
>
> Compared to the baseline methods, it can be seen that UCE totally fails when the number of concepts increases.
> Our AGE method with Oxford-3K vocabulary (AGE-O) outperforms the ESD method in both erasure and preservation performance.
> Compared among the three vocabularies, it can be seen that while AGE-O outperforms other vocabularies in erasure performance,
> it has significant performance drop in preservation performance.
> The AGE method with manually crafted vocabulary (AGE-M) achieves the best trade-off between erasure and preservation performance,
> which has a small drop in erasure performance compared to ESD but a much better preservation performance,
> which is consistent with our analysis in Section 3.2.
>
> | Concept | SD | ESD | UCE | AGE-I | AGE-O | AGE-M |
> |---------|-----|-----|-----|--------|--------|--------|
> | **Erased Concepts** | | | | | | |
> | dog | 99.4 | 11.6 | 0.0 | 35.8 | 11.8 | 7.6 |
> | truck | 99.4 | 23.4 | 0.0 | 14.4 | 9.8 | 6.4 |
> | inst. | 98.6 | 10.8 | 0.0 | 28.8 | 12.8 | 16.0 |
> | build. | 91.2 | 38.8 | 0.0 | 61.2 | 53.0 | 74.8 |
> | elect. | 95.0 | 23.8 | 0.0 | 31.6 | 11.8 | 52.0 |
> | **Similar Concepts** | | | | | | |
> | dog | 100.0 | 94.4 | 0.0 | 99.8 | 92.2 | 99.2 |
> | truck | 100.0 | 79.4 | 0.0 | 99.4 | 78.6 | 96.4 |
> | inst. | 98.8 | 33.4 | 0.0 | 80.4 | 34.8 | 84.8 |
> | build. | 97.4 | 77.0 | 0.0 | 90.6 | 86.6 | 95.2 |
> | elect. | 92.0 | 21.6 | 0.0 | 69.6 | 35.6 | 80.0 |
> | **General Concepts** | | | | | | |
> | mamm. | 99.8 | 98.2 | 0.0 | 99.8 | 98.4 | 100.0 |
> | bird | 99.6 | 87.0 | 0.2 | 98.4 | 86.6 | 97.2 |
> | rept. | 95.6 | 83.2 | 0.0 | 94.4 | 83.8 | 89.2 |
> | insect | 78.6 | 66.4 | 0.0 | 75.8 | 65.0 | 74.4 |
> | fish | 93.8 | 73.0 | 0.0 | 95.8 | 64.6 | 92.8 |
> | veh. | 99.8 | 82.8 | 0.0 | 98.8 | 80.0 | 98.0 |
> | craft | 99.2 | 64.2 | 0.0 | 96.6 | 70.4 | 94.4 |
> | furn. | 96.8 | 64.6 | 0.4 | 83.0 | 72.0 | 80.4 |
> | fruit | 100.0 | 81.6 | 0.0 | 99.8 | 83.8 | 99.2 |
> | obj. | 100.0 | 74.4 | 0.2 | 94.8 | 76.8 | 95.6 |
> | **Metrics** | | | | | | |
> | ESR-1 | 24.2 | 90.6 | 100.0 | 84.7 | 91.2 | 86.1 |
> | ESR-5 | 3.3 | 78.3 | 100.0 | 65.6 | 80.2 | 68.6 |
> | PSR-1 | 83.1 | 56.3 | 0.0 | 74.7 | 57.1 | 73.5 |
> | PSR-5 | 96.8 | 72.1 | 0.1 | 91.8 | 73.9 | 91.8 |

---

> ### Author Response · Authors · 2024-11-25
> **Results on Stable Diffusion v2.1**
>
> **Results on Stable Diffusion v2.1**
>
> We follow the reviewer's suggestion and evaluate the AGE method on Stable Diffusion v2.1. We follow the same experimental settings as in Section 5.1 of the paper. More specifically, we conduct four different experiments, each involving the simultaneous erasure of five classes from the Imagenette dataset while preserving the remaining five classes, generating 500 images per class. While we can successfully deploy the ESD method on Stable Diffusion v2.1, we are unable to do the same for the UCE method because of a implementation issue. All the code has been uploaded to the anonymous GitHub repository.
>
> It can be seen from table below that our method achieves significant improvements over the ESD method in both erasure and preservation performance,
> with a gain of 2.5\% in ESR-5 and 8\% in PSR-5. This indicates the generalizability of our method on different generative models.
>
> We will add the results in the final version of the paper.
>
> | Vocab | ESR-1↑ | ESR-5↑ | PSR-1↑ | PSR-5↑ |
> |-------|--------|---------|---------|---------|
> | SD | 18.28 ± 8.97 | 1.80 ± 0.64 | 81.72 ± 8.97 | 98.20 ± 0.64 |
> | ESD | 91.99 ± 6.35 | 87.83 ± 7.79 | 53.54 ± 8.22 | 75.45 ± 6.43 |
> | AGE | 92.75 ± 6.96 | 90.27 ± 8.73 | 62.45 ± 13.30 | 83.43 ± 9.71 |

---

### Author Response · Authors · 2024-11-25
**Global response**

Dear the reviewers,

We sincerely thank you for your time and effort in reviewing our paper and for providing constructive comments. We appreciate your positive feedback on our work and your acknowledgements on the strengths and contributions made in our paper, which we humbly reiterate below for reference.

Reviewer TuUt: " This method departs from the traditional fixed-target strategy by dynamically selecting optimal target concepts, which is a significant innovation in the field. The modeling of the concept space as a graph to understand the localized impact of concept erasure is a creative and original contribution.... In summary, the paper is a strong contribution to the field, offering original insights and a high-quality, well-executed methodology with significant implications for the future of diffusion models and concept erasure"

Reviewer eC2n: "Clever and intuitive knowledge graph-based approach, Clear and well motivated storyline for the proposed objective, with wide variety of empirical experiments"

We note that the reviewers' raised a shared concern regarding the scalability of our method.
We have addressed this in detail in our rebuttal. In summary, we provided a computational complexity analysis to demonstrate that the optimization problem depends on the batch size (i.e., the number of concepts called per iteration) and the size of the subset of concepts $\mathcal{C}$ rather than the total number of concepts. Furthermore, our implementation employs a dictionary structure to store concept embeddings, enabling efficient updates to target concept embeddings during optimization. To further validate the scalability of our method, we have provided an additional experiment on larger erasure tasks which shows that our method is capable of handling large-scale erasure tasks. All the additional experiments have been uploaded to the anonymous Github repository for reference.

For other concerns raised, we have provided detailed responses in the corresponding sections of the rebuttal. We believe these address all points comprehensively.

If there are any remaining questions or further clarifications needed, we would be delighted to discuss them.

Thank you once again for your constructive feedback and thoughtful review.

The authors

---

### Meta-Review · Area_Chair_EkHB · 2024-12-19

**Metareview:**

The paper proposes AGE - a minimax optimization-based dynamical selection technique for mapping undesirable concepts to target concepts while preserving benign (i.e., desirable) concepts. AGE models the concept space as a graph. It shows that: (a) the effect of concept erasure is localized, and (b) at a local neighborhood level synonymous concepts are not optimal as target erasure.

I agree with majority reviewers that AGE is an original and creative approach toward concept erasure. The motivation is clear and can be agreed upon. Comprehensive experiments have been done to establish the robustness of AGE (along with validation across SOTA generative tasks and architectures). The paper has received credit in terms of clarity.

Certain concerns that if addressed can make the paper stronger:
1. Scalability issues with large concept spaces and computational demands.
2. Dependence on the accuracy of the concept graph and lack of discussion on potential inaccuracies.
3. Limited exploration of generalization to other diffusion or generative models.
4. Insufficient clarity in presenting results and explaining performance gains or shortcomings.
5. Need for more robust evaluation metrics and human evaluations.

**Additional Comments On Reviewer Discussion:**

There has not been any discussion although the authors have made commendable effort in clarification. To summarize the reviews, Reviewer TuUt focuses on the broader contributions and originality, Reviewer eC2n is more skeptical about novelty, and Reviewer BfS9 centers on scalability and specific result interpretations.

---

### Decision · Program_Chairs · 2025-01-22

Accept (Poster)